# GPX1 and RCN1 as New Endoplasmic Reticulum Stress-Related Biomarkers in Multiple Sclerosis Brain Tissue and Their Involvement in the APP-CD74 Pathway: An Integrated Study Combining Machine Learning and Multi-Omics

**DOI:** 10.3390/ijms26136286

**Published:** 2025-06-29

**Authors:** Zhixin Qiao, Yanping Wang, Xiaoru Ma, Xiyu Zhang, Junfeng Wu, Anqi Li, Chao Wang, Xin Xiu, Sifan Zhang, Xiujuan Lang, Xijun Liu, Bo Sun, Hulun Li, Yumei Liu

**Affiliations:** 1Department of Neurobiology, School of Basic Medical Sciences, Harbin Medical University, Harbin 150081, China; 2The Key Laboratory of Preservation of Human Genetic Resources and Disease Control in China, Harbin Medical University, Ministry of Education, Harbin 150081, China; 3The Key Laboratory of Myocardial Ischemia, Harbin Medical University, Ministry of Education, Harbin 150081, China

**Keywords:** multiple sclerosis, endoplasmic reticulum stress, biomarkers, immune infiltration, scRNA-seq, snRNA-seq, neuroinflammation

## Abstract

This study identified 13 endoplasmic reticulum stress (ERS)-related biomarkers associated with multiple sclerosis (MS) through integrated bioinformatics analysis (including weighted gene co-expression network analysis and machine learning algorithms) and single-cell sequencing, combined with validation in an experimental autoimmune encephalomyelitis (EAE) mouse model. Among them, *GPX1*, *RCN1*, and *UBE2D3* exhibited high diagnostic value (AUC > 0.7, *p* < 0.05), and the diagnostic potential of *GPX1* and *RCN1* was confirmed in the animal model. The study found that memory B cells, plasma cells, neutrophils, and M1 macrophages were significantly increased in MS patients, while naive B cells and activated NK cells decreased. Consensus clustering based on key ERS-related genes divided MS patients into two subtypes. Single-cell sequencing showed that microglia and pericytes were the cell types with the highest expression of key ERS-related genes, and the APP-CD74 pathway was enhanced in the brain tissue of MS patients. Mendelian randomization analysis suggested that *GPX1* plays a protective role in MS. These findings reveal the mechanisms of ERS-related biomarkers in MS and provide potential targets for diagnosis and treatment.

## 1. Introduction

Multiple sclerosis (MS) is one of the most common autoimmune diseases of the central nervous system (CNS), characterized by chronic inflammation, demyelination, and progressive neurodegeneration [1]. Globally, the disease affects approximately 2.5 million people and exhibits geographical variations. However, its etiology remains poorly understood, involving a complex interplay of genetic susceptibility, environmental factors, and immune dysregulation [2,3,4]. Despite advances in immunomodulatory therapies, there is no definitive cure, highlighting the need for a deeper understanding of MS pathophysiology to identify novel biomarkers and therapeutic targets [5].

One emerging pathological mechanism in MS is endoplasmic reticulum stress (ERS), which plays a crucial role in protein homeostasis and immune regulation. ERS triggers the unfolded protein response (UPR), a cellular protective mechanism [6]. However, under chronic inflammatory conditions, sustained ERS activation can induce apoptosis and neuroinflammation, further disrupting immune homeostasis in the CNS [7,8]. Previous studies have demonstrated that targeting ERS pathways can mitigate neuroinflammation in neurodegenerative diseases such as Alzheimer’s disease, yet the role of ERS in MS remains largely unexplored [9]. Identifying ERS-related differentially expressed genes (ERS-DEGs) in MS could provide valuable insights into disease pathogenesis and facilitate the development of novel biomarkers for diagnosis and treatment.

Recent advancements in high-throughput transcriptomic analysis and bioinformatics have enabled systematic exploration of disease mechanisms at the molecular level [10]. Multi-omics approaches, including weighted gene co-expression network analysis (WGCNA), single-cell RNA sequencing (scRNA-seq), and immune infiltration analysis, have provided powerful tools for uncovering disease-specific gene signatures. Against this backdrop, several studies have utilized bioinformatics methods to investigate endoplasmic reticulum stress in acute myeloid leukemia and successfully constructed prognostic models [11].

To elucidate the molecular mechanisms of ERS in the progression of MS and its association with immune infiltration, we employed a combination of bioinformatics and experimental validation to investigate the expression of ERS-related genes in MS brain tissue in this study. Specifically, we: 1. Identified ERS-DEGs by integrating transcriptomic datasets from the Gene Expression Omnibus (GEO). 2. Explored the relationship between ERS and immune infiltration in MS using computational analysis and single-cell sequencing data. Additionally, through cell communication analysis of cells with high Key ERS-DEGs scores in MS, we identified the specific cellular interaction mechanisms related to ERS. 3. Conducted consensus clustering of MS using the selected Key ERS-DEGs to obtain molecular subtypes and predict relevant drugs. 4. Validated key ERS-DEGs through experimental autoimmune encephalomyelitis (EAE), an animal model of MS. 5. Analyzed the changes in CD4+ T cells, CD19+ B cells, and neutrophils in EAE brain tissue using flow cytometry. 6. Investigated the potential causal associations between key ERS-DEGs and MS using epidemiological research methods.

Our findings provide new insights into the immunopathological mechanisms of MS, enriching the pool of potential diagnostic biomarkers. They highlight ERS as a potential diagnostic and therapeutic target for the disease, offering new ideas for subsequent drug development and the optimization of clinical diagnostic strategies. The flowchart illustrates the overall research design and workflow (Figure 1).

## 2. Result

### 2.1. Identification of DEGs Between Brain Tissues of Multiple Sclerosis Patients and Controls

To ensure the comparability of gene expression data among samples, we performed normalization and batch effect removal on the GSE131282 and GSE135511 datasets (Figure 2A). Through WGCNA, after removing two outlier samples, we selected β = 13 (R^2^ = 0.85) as the “soft” threshold based on average connectivity (Appendix A). The blue module was ultimately identified as the one most strongly associated with MS, and a strong correlation was found between module membership and gene significance within this module (Figure 2B,C, Appendix A).

The volcano plot analysis identified a total of 1932 DEGs in the brain tissues of MS patients compared to the control group, with the heatmap displaying the top 20 upregulated and downregulated DEGs (Figure 3A,B). Gene Ontology (GO) analysis revealed that DEGs were mainly enriched in biological processes (BP) such as negative regulation of apoptosis and Ras protein signal transduction. Regarding molecular function (MF), DEGs were closely related to protein binding and G protein activity (Figure 3C). KEGG analysis showed that DEGs were primarily enriched in metabolic pathways, autophagy, and the cAMP signaling pathway (Figure 3D).

Further analysis revealed that after intersecting DEGs with the genes in the MS-related module identified by WGCNA, 324 common genes were obtained. After intersecting these 324 common genes with 785 ERS-related genes, 13 ERS-DEGs were ultimately identified (Figure 3E). These results reveal the biological phenomena potentially involved in the pathogenesis of MS, including the regulation of apoptosis, abnormal signal transduction, and differential expression of ERS-related genes.

### 2.2. Identification and Validation of Key ERS-DEGs

We selected 13 ERS-DEGs from the merged dataset to identify potential key genes. We employed a combination of machine learning methods, including the linear model LASSO regression and the tree-based model Boruta, to identify key genes (Appendix A). Through LASSO regression modeling, we identified 5 key genes. The Boruta algorithm was used to rank the importance of genes in the model, and we selected four genes with an importance score greater than 10 (Figure 4A,B). By intersecting the genes identified by both models, we ultimately determined 4 key ERS-DEGs (Figure 4C).

We evaluated the diagnostic value of these key ERS-DEGs using ROC curves constructed from the external dataset GSE108000. *GPX1*, *RCN1*, and *UBE2D3* exhibited AUC values above 0.7 in both datasets, indicating their potential as biomarkers for multiple sclerosis (Figure 5A,B).

Next, we analyzed the expression levels of these three genes in both the merged and external datasets. The expression levels of *GPX1*, *RCN1*, and *UBE2D3* were significantly higher in MS samples than in controls (*p* < 0.05) (Figure 5C,D).

We further validated the expression of these three key ERS-DEGs in an animal model of the disease by establishing an EAE model and using RT-PCR. The results showed that *GPX1* and *RCN1* were significantly upregulated in the brains of EAE mice compared to controls (Figure 5E). These findings are generally consistent with the expression patterns observed in our merged dataset and the external dataset.

### 2.3. Immune Infiltration Analysis in MS

To investigate the neuroinflammatory characteristics of multiple sclerosis (MS), we employed the CIBERSORT method to analyze immune cell infiltration in MS patients and controls. The MS group exhibited significantly higher proportions of memory B cells, plasma cells, M1 macrophages, naive CD4+ T cells, and neutrophils (*p* < 0.05), whereas the control group had higher proportions of naive B cells, activated natural killer (NK) cells, and activated dendritic cells (*p* < 0.05) (Figure 6A). A stacked bar chart illustrated the distribution of immune cell proportions across samples (Appendix A). Correlation analysis showed that plasma cells were negatively correlated with naive B cells (correlation coefficient > 0.5), indicating a potential interaction between these cell types (Appendix A). Further analysis revealed that key ERS-DEGs were significantly correlated with various immune cell types (Figure 6B,C), suggesting that these genes may influence MS pathogenesis through immune cell regulation.

To validate these findings, we used flow cytometry to analyze the proportions of CD19+ B cells, CD4+ T cells, and neutrophils in the brain tissues of EAE mice. Compared with controls, EAE mice showed significant increases in the proportions of CD4+ T cells, CD19+ B cells, and neutrophils (Figure 7A–C and Figure 8A,B). These results underscore the critical role of immune cell infiltration in MS pathology and highlight potential therapeutic targets for immune-based interventions.

### 2.4. Unsupervised Clustering and Analysis of MS

To determine whether two key ERS-DEGs can distinguish MS samples, we analyzed 57 samples using the consensus clustering algorithm. Based on the consensus cumulative distribution function, we divided the samples into two groups (Figure 9A,B): Group A (n = 27) and Group B (n = 30). Differential analysis revealed that the expression levels of *GPX1* and *RCN1* were lower in Group A compared to Group B (Figure 9C,D). A volcano plot identified a total of 53 DEGs, including 36 upregulated and 17 downregulated genes (Figure 9E).

Based on the highly expressed genes in each group, we predicted potential drugs targeting these subtypes. In Group A, PAREGORIC targets the *GRM1* and *OPRK1* genes, while METFORMIN targets the *NBEA* and *PRKAA2* genes (Appendix A). Additionally, CIBERSORT analysis showed that Group A had a higher proportion of naive B cells, whereas Group B had a higher proportion of CD8+ T cells (*p* < 0.05) (Figure 9F).

### 2.5. Expression of Key ERS-DEGs in Single-Cell Data

To elucidate the expression profiles of key ERS-DEGs in MS brain tissue, we examined the UMAP plot from the GSE118257 snRNA-seq data, which identified the distribution of 12 cell types, including astrocytes, endothelial cells, ependymal cells, fibroblasts, microglia, neural stem cells, neurons, oligodendrocytes, oligodendrocyte precursor cells (OPCs), pericytes, stromal cells, and T cells (Figure 10A). A stacked bar chart highlighted the differences in cell proportions between the MS and control groups (Figure 10B). The expression patterns of two key ERS-DEGs, *GPX1* and *RCN1*, revealed that *GPX1* is predominantly expressed in endothelial cells, microglia, pericytes, stromal cells, and T cells, while *RCN1* is primarily expressed in astrocytes, endothelial cells, ependymal cells, OPCs, stromal cells, and pericytes (Figure 10C). In the MS group, *GPX1* was downregulated in pericytes but upregulated in microglia and fibroblasts. Conversely, *RCN1* was downregulated in OPCs and endothelial cells but upregulated in pericytes (Appendix A).

Similarly, the UMAP plot from the GSE199460 scRNA-seq data from EAE mouse brain tissue identified 17 cell types, including astrocytes, B cells, cDC2, choroid plexus cells, endothelial cells, ependymal cells, fibroblasts, M1 macrophages, M2 macrophages, microglia, neurons, neutrophils, neural progenitor cells (NPCs), OPCs, pericytes, and T cells (Figure 11A). A stacked bar chart highlighted the differences in cell proportions between the EAE and control groups (Figure 11B). Further analysis showed that *GPX1* is mainly expressed in B cells, cDC2, choroid plexus cells, endothelial cells, fibroblasts, M1 macrophages, M2 macrophages, microglia, neutrophils, OPCs, and pericytes. In contrast, *RCN1* is primarily expressed in astrocytes, choroid plexus cells, fibroblasts, ependymal cells, OPCs, pericytes, and T cells (Figure 11C). In the EAE group, *GPX1* was significantly downregulated in cDC2 cells but upregulated in oligodendrocytes, NPCs, neurons, microglia, M1 macrophages, fibroblasts, choroid plexus cells, endothelial cells, and astrocytes. *RCN1* was significantly upregulated in oligodendrocytes, NPCs, neurons, microglia, M2 macrophages, B cells, and astrocytes (Appendix A).

### 2.6. Analysis of Cell Communication in MS

To explore which cells are primarily responsible for the function of key ERS-DEGs in the cell communication network, we used the “AddModuleScore” function in the Seurat package to score cell types based on key ERS-DEGs. We found that microglia and pericytes had the highest scores in MS (Figure 12A). The cell communication analysis showed that the number and intensity of cell communication pathways were significantly higher in the MS group than in the control group (Appendix A). The chord diagram revealed that microglia mainly communicated with oligodendrocytes, OPCs, astrocytes, and ependymal cells, while pericytes had extensive signal interactions with multiple cell types (Figure 12B). The heatmap further displayed the signal output and input of each pathway among different cells (Figure 12C,D).

### 2.7. Conducting Separate Analyses of Cell Communication via the APP-CD74 Pathway in MS and the EAE Mouse Model

To identify differentially expressed pathways between MS and controls, we compared the cell communication analysis results between the MS and control groups. Since microglia and pericytes are the two cell types with the highest scores based on key ERS-DEGs, we used a dot plot to illustrate the pathway differences between these two cell types across the two groups. The size of the dot represents the significance of the difference, while the color of the dot represents the intensity of the difference. We found that in the APP-CD74 pathway, microglia act as signal receivers, while pericytes act as signal senders, and this pathway is significantly enhanced in MS (Figure 13A,B), as indicated by a red box.

Next, we conducted a separate analysis of the APP-CD74 pathway. The horizontal bar chart shows that APP-CD74 is the core ligand-receptor pair in this pathway (Appendix A). The hierarchical plot indicates that microglia are important signal receivers in the APP-CD74 pathway (Figure 13C), a finding supported by the dot plot, heatmap, and chord diagram (Appendix A). The network centrality score reveals the roles of various cells in the APP-CD74 pathway (Figure 13D). Interestingly, microglia not only serve as important signal receivers but also exert significant influence within this pathway. Finally, the violin plot displays the expression of *APP* and *CD74* among different cell types. Microglia primarily express CD74, while pericytes mainly express APP in MS (Figure 13E). We additionally obtained the immunohistochemical results of APP in brain tissue from the Human Protein Atlas (HPA) database to observe its expression (Appendix A).

To validate the APP-CD74 pathway in the MS mouse model, we used scRNA-seq data from the EAE model. A dot plot showed interactions between microglia (signal receivers) and pericytes (signal senders) with other cells via the APP-CD74 pathway. Results indicated that this pathway is significantly enhanced between pericytes and microglia in EAE, consistent with MS findings (Figure 14A–D). Bar and chord diagrams showed increased cell communication number and intensity in EAE (Appendix A). The horizontal bar and hierarchical plot charts highlighted APP-CD74 as the core ligand-receptor pair, with microglia as key signal receivers (Figure 14E, Appendix A). The heatmap also supported these conclusions (Appendix A). Network centrality scores further confirmed these roles (Figure 14F). A violin plot displayed *APP* and *CD74* expression across different cells (Figure 14G). We performed immunohistochemistry to localize APP and CD74 in brain tissue from control and EAE mice. We found that in EAE mice, APP expression was enhanced around blood vessels in the brain. Meanwhile, we observed increased expression of CD74 in the brain tissue of EAE mice, with an irregular shape characteristic of microglia (Appendix A). Immunofluorescence results showed co-localization between APP and the pericyte marker PDGFRβ, as well as between CD74 and the microglial marker IBA-1, which was enhanced in the brain tissue of EAE mice (Appendix A). This further supports the results of the cell communication analysis.

### 2.8. Functional Roles of Microglia and Pericytes via the APP-CD74 Pathway in MS

To further investigate the roles of microglia and pericytes through the APP-CD74 pathway, as well as the relationship between key ERS-DEGs and the APP-CD74 ligand–receptor pair, we analyzed gene expression differences between microglia and pericytes in MS. The results showed that *APP* and *RCN1* are highly expressed in pericytes, while *CD74* and *GPX1* are highly expressed in microglia (Figure 15A–E). Enrichment analysis indicated that pericytes primarily function in antioxidant detoxification and hydrogen peroxide metabolism (Figure 15F).

Given that microglia represent a high proportion of cells in the dataset, we divided them into three subtypes (Micro1, Micro2, and Micro3) and analyzed the highly expressed genes in each subtype. Functional enrichment analysis revealed that the Micro3 subtype is associated with innate immunity and positive regulation of tumor necrosis factor production, indicating its involvement in inflammation and immune responses (Figure 16A–D). Additionally, the proportion of the Micro3 subtype significantly increased in the MS group (Figure 16E).

Further analysis showed that *CD74* and *GPX1* expression were significantly elevated in the Micro3 subtype (Figure 16F–I). Therefore, we propose that *GPX1* may be related to the function of the Micro3 subtype in MS, while *RCN1* is associated with the function of pericytes. These changes may be related to the APP-CD74 pathway and influenced by key ERS-DEGs.

### 2.9. MR Analysis Between Key ERS-DEGs and MS

To interpret our identified potential biomarkers at a broader epidemiological level, we downloaded data from the OpenGWAS database based on the diseases and gene IDs we retrieved and processed the VCF files to obtain data suitable for further analysis (Appendix A).

Based on our key ERS-DEGs, we used the TwoSampleMR framework to conduct a two-sample Mendelian randomization (MR) analysis for the MS-specific single-nucleotide polymorphisms (SNPs) and the expression quantitative trait loci (eQTL) data of the two key ERS-DEGs, one by one. The analysis showed that the inverse-variance weighted (IVW) test indicated a negative correlation between *GPX1* and MS (*GPX1*: OR = 0.876, *p* < 0.05) (Figure 17), while no significant causal relationship was found between *RCN1* and MS (Figure 18). In summary, we concluded that among the two key ERS-DEGs, *GPX1* has a potential causal association with multiple sclerosis (MS) and plays a protective role in MS.

## 3. Discussion

Weighted gene co-expression network analysis (WGCNA) is a systems biology method that can reveal the correlation of gene expression and identify gene modules associated with phenotypes [12]. After obtaining the most disease-related modules through WGCNA and then combining them with DEGs, this is a more efficient method than performing WGCNA only among DEGs after screening for DEGs [13]. In this study, we used this method to identify ERS-DEGs most relevant to MS brain tissue and obtained key ERS-DEGs using the LASSO + Boruta combined machine learning approach. To evaluate the value of these key ERS-DEGs as potential biomarkers, we validated them using external datasets and disease animal models. Ultimately, we identified *GPX1* and *RCN1* as potential biomarkers and analyzed the association between ERS and immune infiltration, as well as the impact of this association on MS, based on these biomarkers.

MS is a multifactorial chronic inflammatory demyelinating disease of the central nervous system, with unclear specific etiology, and is characterized by immune infiltration and neurodegenerative phenomena [14]. Given its long disease course and lack of early specific clinical manifestations, MS is difficult to diagnose [15]. Therefore, investigating the pathogenesis of MS to identify biomarkers with diagnostic value is crucial for optimizing therapeutic strategies and developing new treatments.

GPX1 is a member of the glutathione peroxidase family. It reduces intracellular ROS accumulation through its antioxidant function, thereby alleviating protein folding stress in the endoplasmic reticulum and mitigating ERS. It is a potential therapeutic target for several diseases. Previous studies have shown that GPX1 regulates oxidative stress and ERS-related apoptosis in Alzheimer’s disease and Parkinson’s disease [16,17,18]. *RCN1* is a member of the CREC family. During ERS, it is activated by NF-κB and protects cells from apoptosis by maintaining calcium homeostasis and inhibiting the PERK-CHOP pathway [19]. Studies have shown that RCN1, as an ER-resident protein, promotes cell survival during ERS [20].

Immune infiltration is closely related to multiple sclerosis (MS). Analysis using the CIBERSORT algorithm [21] revealed that, compared with the control group, the proportions of memory B cells, plasma cells, M1 macrophages, naive CD4^+^ T cells, and neutrophils were significantly higher in the MS group, while the proportions of naive B cells, activated NK cells, and activated dendritic cells were significantly lower. These cells play different roles in MS progression: Memory B cells and plasma cells promote MS progression by enhancing the proliferation of autoreactive T cells and producing autoantibodies [22,23]; M1 macrophages and neutrophils exacerbate neuroinflammation by releasing inflammatory mediators [24,25,26]; whereas naive B cells, NK cells, and dendritic cells are involved in immune responses or exert neuroprotective effects [27,28,29]. Endoplasmic reticulum stress (ERS) promotes the production of inflammatory cytokines by activating the NF-κB and MAPK signaling pathways and recruits immune cells by secreting cytokines and chemokines, thereby influencing tissue inflammatory responses [30,31]. Correlation analysis showed that the two key ERS-related genes were positively correlated with pro-inflammatory cells (such as neutrophils, memory B cells, M1 macrophages, and plasma cells), further elucidating the link between ERS and pro-inflammatory immune cell infiltration.

To further explore how key ERS-DEGs specifically affect MS, we first used the AddModuleScore function to score cells in snRNA-seq and found that microglia and pericytes were the two types of cells with the highest key ERS-DEGs scores. Previous studies have shown that during the progression of neurodegenerative diseases, microglia and pericytes undergo endoplasmic reticulum stress due to the accumulation of misfolded proteins, thereby affecting disease progression [32,33,34]. Based on the above findings, we constructed and compared the cell communication networks of microglia and pericytes between the MS group and the control group through cell communication analysis. This method provides more biological insights [35]. The comparison results show that in MS, the APP-CD74 pathway between microglia and pericytes is significantly enhanced, and this enhancement also appears in the brain tissue scRNA-seq dataset from EAE mice. Pericytes, a major component of the brain vasculature, widely express amyloid precursor protein (APP) in the central nervous system (CNS) vascular system. In proliferative diabetic retinopathy (PDR), pericytes exhibit global communication through the APP signaling pathway, with APP-CD74 being the major ligand–receptor pair and microglia serving as one of the key receiving cells [36]. The enzymatic cleavage of APP generates Aβ, and both APP and Aβ are in dynamic equilibrium within the CNS’s cerebral vascular system. In AD, Aβ and pericytes have been shown to co-localize [37,38], suggesting that pericytes and microglia may be involved in maintaining the dynamic balance between APP and Aβ. 

Based on the functional interactions revealed by single-cell transcriptomics, we further referred to the immunohistochemistry (IHC) data in the Human Protein Atlas (HPA) database. The HPA is a public database of proteins, containing a wealth of data on the differential expression of proteins in different tissues. Given that CD74 is one of the well-recognized markers of microglial activation and has been revealed in MS [39], we obtained the IHC image data of APP from the HPA database. The results showed that APP was localized in the vascular-related areas of inflamed brain tissue, consistent with the distribution of pericytes. Although this evidence is indirect and not from MS-specific tissues, neuroinflammation is one of the main pathological phenomena in many neurological diseases, including MS and AD. Therefore, this also supports our conclusion from another perspective. The immunohistochemical results from our analysis of brain tissue sections from EAE mice confirmed the findings from cell communication analysis, which showed that APP expression was increased in pericytes of the brain vascular unit in EAE mice, while CD74 expression was increased in microglia. Co-localization analysis by immunofluorescence further supported these conclusions.

Based on the above results and the immune phenotypes of microglia in MS [40], we further conducted a subpopulation analysis of microglia. In the APP-CD74 pathway, microglia, as important receptors, express *CD74*. We found that in the subpopulation of microglia with higher *CD74* expression, *GPX1* is also highly expressed. Because the expression of NF-κB during inflammation can increase *GPX1* expression, and in MS, the expression of NF-κB does indeed increase, and the deposition of β-amyloid can exacerbate endoplasmic reticulum stress. However, the activation of the APP-CD74 pathway can reduce the deposition of β-amyloid [41,42,43,44]. Therefore, *GPX1* may function by affecting the ability of microglia to handle β-amyloid through the APP-CD74 pathway, thereby further alleviating endoplasmic reticulum stress. As one of the signal senders of the APP-CD74 pathway in MS, although pericytes are a relatively small cell population, they play an important role in maintaining the blood–brain barrier [45]. The deposition of β-amyloid can affect the blood–brain barrier by influencing the state of pericytes [46]. In MS, the expression of *RCN1* in pericytes is increased. Therefore, although the expression level of *RCN1* itself is not high, its function of promoting cell survival during endoplasmic reticulum stress may be reflected in pericytes.

On the basis of the two key ERS-DEGs, we identified two clusters through consensus clustering, with significantly higher expression of these genes in Cluster B than in Cluster A. We searched for potential drugs targeting the differentially expressed genes in these two molecular subtypes, including PAREGORIC and METFORMIN (a drug used for type 2 diabetes and beneficial for immune cells in MS) [47]. Despite the limitations of subtyping based on a small number of genes, the differences between these subtypes provide new insights into the potential use of biomarkers. CIBERSORT analysis showed that compared with Cluster A, the proportion of CD8+ T cells in Cluster B was significantly higher. This difference in immune cell infiltration further revealed that key ERS-DEGs may have a potential regulatory effect on immune cells. For the *GRM1* and *OPRK1* genes targeted by PAREGORIC, they are glutamate receptor and opioid receptor, respectively. Studies have pointed out that they are potential research genes for regulating immune cells [48,49]. For the *NBEA* and *PRKAA2* genes targeted by METFORMIN, they are Neurobeachin and Protein kinase AMP-activated catalytic subunit alpha 2, respectively. Among them, *PRKAA2* is closely related to ferroptosis, and the regulation of immune cells by ferroptosis has been widely studied [50]. However, how these genes further affect immune infiltration between subtypes requires more in-depth research in the future.

To better elucidate the relationship between these two key ERS-DEGs and MS, we employed the Mendelian randomization (MR) method, using the two genes as exposures and MS as the outcome for a two-sample MR analysis. The MR analysis suggested a potential protective association between *GPX1* and MS risk, although further experimental validation is required to confirm causality. Previous studies have demonstrated that *GPX1* is a known therapeutic target of curcumin, which can be used to treat EAE [51]. This finding supports our results.

Although our findings are primarily based on machine learning and multi-omics analysis, additional experimental validation of the conclusions supports the conclusions obtained from our bioinformatics analysis and further affirms the effectiveness and efficiency of the workflow. This not only lays a solid foundation for future, more in-depth mechanistic studies but also provides a reference for an efficient integrated bioinformatics workflow. In the future, we will employ techniques such as spatial transcriptomics, proximity ligation assays (PLA), and fluorescence dynamic imaging to confirm the spatial localization and functional interactions between APP, CD74, and related ERS regulators.

## 4. Materials and Methods

### 4.1. Data Acquisition

The gene expression data for MS and control samples were obtained from the Gene Expression Omnibus (GEO) database. The tissue selection was brain tissue, including RNA-seq data from GSE131282 (42 MS samples and 37 control samples) [52], RNA-seq data from GSE135511 (20 MS samples and 10 control samples) [53], RNA-seq data from GSE108000 (30 MS samples and 10 control samples) [54], snRNA-seq data from GSE118257 (15 MS samples and 5 control samples) [55], and scRNA-seq data from GSE199460 (3 EAE samples and 3 control samples) [56]. For bulk RNA-seq data, the expression matrix was normalized and background-corrected using the normalize-betweenarrays function of the R package limma (version 3.60.6). After filtering out empty probes, the data were annotated according to the chip probe–gene correspondence. Finally, a normalized dataset with complete annotation was obtained, which can be used for further analysis. If multiple probes corresponded to the same gene expression value, the average value was selected as the expression level for that gene. The endoplasmic reticulum stress (ERS)-related gene set was derived from the previous literature [57], consisting of a total of 787 genes (Appendix A).

### 4.2. Weighted Gene Co-Expression Network Analysis

We first used the R package “sva” (version 3.52.0) to remove batch effects from GSE131282 and GSE135511 and then merged the datasets. Subsequently, the R package “WGCNA” (version 1.73) was used to identify potential functional modules. WGCNA was performed on the gene expression profiles and clinical data of the merged dataset. All samples were clustered based on average linkage and Pearson correlation values. A suitable soft threshold β was selected by converting the adjacency matrix into a topological overlap matrix (TOM). A hierarchical clustering tree was constructed, with the branches of the tree representing gene modules (the minimum number of genes per module was set at 100). The correlation between the resulting modules and clinical traits was analyzed, and modules significantly associated with the traits were selected. Scatter plots were constructed to show the correlation between gene significance and module membership.

### 4.3. Identification of Differentially Expressed Genes (DEGs)

We used the R package “limma” (version 3.60.6) to extract DEGs from the merged dataset. A threshold of |log2(FC)| > 0.5 and an adjusted *p* < 0.05 was set for identifying DEGs, and the differential analysis was visualized using R packages. Additionally, Gene Ontology (GO) enrichment analysis and Kyoto Encyclopedia of Genes and Genomes (KEGG) pathway analysis were performed using the DAVID tool [58,59], and the enriched functions and pathways with “*p* < 0.05” were retained and visualized using the R package “ggplot2” (version 3.5.1).

### 4.4. Identification and Validation of ERS-DEGs Associated with Clinical Traits in MS

Using the R package “VennDiagram” (version 1.7.3), we intersected the DEGs with the modules selected by WGCNA and then with the ERS-related gene set to obtain ERS-DEGs. To identify stable and reproducible key ERS-DEGs in MS, we combined the least absolute shrinkage and selection operator (LASSO) and Boruta methods using the R packages “glmnet” (version 4.1-8) and “Boruta” (version 8.0.0). These two methods, as machine learning strategies, have been used in recent years to screen features from high-throughput sequencing data [60]. Based on the key ERS-DEGs, we used the R package “pROC” (version 1.18.5) to generate receiver operating characteristic (ROC) curves in the merged dataset and the external dataset GSE108000 to evaluate and validate the diagnostic value of the biomarkers. Key ERS-DEGs with an area under the curve (AUC) greater than 0.7 in both the merged and external datasets were selected for further analysis.

### 4.5. Immune Infiltration Analysis in Multiple Sclerosis Using CIBERSORT

The R package “CIBERSORT” (version 0.1.0) was used to quantify immune infiltration in MS brain tissue. This is a gene expression-based deconvolution algorithm that uses a leukocyte gene signature matrix. Consequently, the output was directly integrated to generate the entire matrix of immune cell components. We compared the distribution of immune cell infiltration between MS and controls. CIBERSORT was employed to calculate the proportions of immune cells to approximate the landscape of immune infiltration. Additionally, we performed Spearman correlation analysis with a significance threshold of *p*-value < 0.05 to assess the relationships between key ERS-DEGs and immune cell infiltration in MS.

### 4.6. Subclustering Analysis Using Key ERS-DEGs

We performed hierarchical clustering analysis on 57 MS samples using the R package “ConsensusClusterPlus” (version 1.70.0), with the gene expression profiles of key ERS-DEGs as input. Subsequently, we compared the differences in immune infiltration between subtypes using CIBERSORT. The R package “limma” (version 3.60.6) was employed to identify differentially expressed genes (DEGs) between subtypes, with a threshold of |log2(FC)| > 0.5 and an adjusted *p*-value < 0.05 for DEG identification. DGIdb (https://dgidb.org/, accessed on 20 February 2025) was used to identify drugs targeting the highly expressed genes in each subtype, retaining those with a regulatory approval status of “Approved”. Finally, we used Cytoscape software (version 3.10.1) to generate the association network diagram.

### 4.7. Processing and Clustering of Single-Nucleus RNA and Single-Cell RNA Data

We processed the GSE118257 and GSE199460 datasets using the R package “Seurat” version 5.1.0 [61]. For quality control, cells with fewer than 200 genes, mitochondrial counts greater than 5%, and genes expressed in fewer than three cells were removed. For subsequent analysis, a resolution of 0.5 was chosen for GSE118257, and a resolution of 0.3 was chosen for GSE199460. The FindAllMarkers function was then employed to identify marker genes for each cluster, with parameters set as (logfc.threshold = 0.25, showing only upregulated genes). Finally, the top 20 genes for each cluster were selected and annotated in combination with CellMarker 2.0 [62] and the original articles of GSE118257 and GSE199460. The expression of key ERS-DEGs in different cell types was visualized using the DotPlot and VlnPlot functions. The cell clustering map was visualized using the “SCpubr” package (version 2.0.2) [63].

### 4.8. Scoring of Key ERS-DEGs in snRNA-seq

We used the AddModuleScore function in the Seurat package to score key ERS-DEGs for each cell type in snRNA-seq. The specific process involves using the key ERS-DEGs as a gene list to evaluate their expression in each cell type.

### 4.9. Cell Communication Analysis

Based on the CellChatDB database, we used the R package “CellChat” (version 1.6.1) [64] to analyze cell communication in both snRNA-seq and scRNA-seq. We set the number of threads to 4 and used the “identifyOverExpressedGenes” function to find overexpressed genes. The “identifyOverExpressedInteractions” function was then used to obtain the cell communication results between cells. Finally, we projected the cell communication results onto the interaction network by setting “PPI.human” or “PPI.mouse”. After obtaining the probability of cell communication using the “computeCommunProb” function, we filtered the cell communication network by setting the threshold min.cell = 10 cells. Subsequently, we selected the top two cell subtypes with the highest key ERS-DEGs scores and used the “compareInteractions” function to obtain their communication networks, respectively, as signal receivers and senders. We compared the cell communication networks between the MS group and the control group and used the “netVisual_bubble” function to draw bubble plots to identify significantly different pathways.

### 4.10. Collection of GWAS Data and Two-Sample Mendelian Randomization Analysis

GWAS data for exposures and outcomes were downloaded from the IEU OpenGWAS Project database. The eQTL data for genes were obtained by retrieving ENSG IDs from the NCBI database (National Center for Biotechnology Information) and then acquiring the eQTL data from the OpenGWAS database. For MS, data were directly searched using “multiple sclerosis” in OpenGWAS. The VCF data were uniformly processed using the R package “gwasglue” (version 0.0.0.9000), and visualization was performed using the R package “CMplot” (version 4.5.1). Subsequently, the R packages “VariantAnnotation” (version 1.52.0) and “TwoSampleMR” (version 0.6.7) were used for two-sample Mendelian randomization analysis. The filtering criteria for exposure SNPs were set as pval.exposure < 0.01, clump_kb = 10,000, clump_r2 = 0.1, and clump_p = 1.

### 4.11. Immunohistochemistry and Immunofluorescence

To verify the expression of biomarkers and related genes in brain tissue, we examined the immunohistochemical staining data of APP in both inflammatory and normal brain tissue samples. The images are from the Human Protein Atlas (https://www.proteinatlas.org, accessed on 20 May 2025).We also performed immunohistochemical and immunofluorescence staining on frozen sections of brain tissue from control and EAE mice. After removing the mouse brain, it was fixed with 4% paraformaldehyde and dehydrated with 15% and 30% sucrose solutions before sectioning. The immunohistochemical staining process was as follows: after washing with 1% PBS solution, the sections were fixed with cold acetone for 15 minutes, followed by treatment with 3% hydrogen peroxide to remove endogenous peroxidase. After blocking with 0.5% BSA for 1 hour, the sections were incubated with mouse anti-APP (1:500, Invitrogen, Carlsbad, CA, USA, 14-9749-82) and rabbit anti-CD74 (1:200, abcam, Cambridge, UK, ab289885) at 4°C for 12–16 hours. Then, HRP-conjugated goat anti-mouse (1:300, Invitrogen, Carlsbad, CA, USA) and goat anti-rabbit (1:300, Invitrogen, Carlsbad, CA, USA) were applied and incubated at 37°C for 1 hour. Finally, the sections were stained with DAB chromogenic solution, and the nuclei were counterstained with hematoxylin and observed under a microscope. The immunofluorescence staining process was as follows: after washing with 1% PBS solution, the sections were fixed with cold acetone for 15 minutes. After blocking with 0.5% BSA for 1 hour, the sections were incubated with mouse anti-APP (1:300), rabbit anti-CD74 (1:100), rabbit anti-PDGFRβ (1:100, abmart, Shanghai, China, T55135S), and goat anti-IBA-1 (1:100, abmart, Shanghai, China, PGG010S) at 4°C for 12–16 hours. Then, fluorochrome-conjugated donkey anti-mouse (1:200, Invitrogen, Carlsbad, CA, USA), donkey anti-rabbit (1:200, Invitrogen, Carlsbad, CA, USA), and donkey anti-goat (1:200, Invitrogen, Carlsbad, CA, USA) were applied and incubated at 37°C for 1 hour. The nuclei were counterstained with DAPI at 0.5 μg/ml and observed under a fluorescence microscope.

### 4.12. Establishment of EAE Model

All experiments were approved by the Ethics Committee of Harbin Medical University. Female C57BL/6 mice, aged 6–8 weeks and weighing 16–18 g, were maintained in a 12 h light–dark cycle. The experimental group was injected subcutaneously with an emulsion containing MOG_35-55_ peptide (Sigma, St. Louis, MO, USA) and complete Freund’s adjuvant on both sides of the spine. On the day of immunization and the following day, the mice were given two intravenous tail vein injections of pertussis toxin (Sigma, St. Louis, MO, USA) at a dose of 400 ng. The control group received only the emulsion without MOG_35-55_ peptide and did not undergo further treatment. The mice were euthanized on day 16 after immunization.

### 4.13. RNA Extraction and qRT-PCR

Total RNA was extracted using Trizol reagent, and cDNA was synthesized using the PrimeScript™ RT reagent kit (Takara, Shiga, Japan). Real-time quantitative PCR analysis was performed using the Universal SYBR qPCR Detection Kit (Beyotime, Shanghai, China). The levels of target mRNA were normalized to the mRNA level of glyceraldehyde-3-phosphate dehydrogenase (GAPDH). The primer sequences are provided in Appendix A. The relative mRNA expression levels were calculated using the 2^−ΔΔCt^ method.

### 4.14. Flow Cytometry Analysis

In the experiments, mice were euthanized under deep isoflurane anesthesia and then perfused with cold 1× PBS at 4°C via cardiac puncture to remove white blood cells from the vasculature. The dissected brain tissue was homogenized in Hank’s balanced salt solution using a tissue homogenizer and then filtered through a 70 μm cell strainer. After centrifugation, the homogenate pellet was resuspended in 40% isotonic Percoll solution and gently layered over 70% isotonic Percoll solution. A Percoll density gradient was formed by centrifugation at 500× *g* for 30 min. The mononuclear layer was collected from the interface of the Percoll solution. After washing with 1× PBS, cells were stained with anti-CD11b (Biolegend, San Diego, CA, USA), anti-Ly6C (Biolegend, San Diego, CA, USA), and anti-Ly6G (Biolegend, CA, USA) to detect the proportion of neutrophils; they were also stained with anti-CD4 (Biolegend, CA, USA) and anti-CD19 (Biolegend, San Diego, CA, USA) to detect the proportion of CD4+ T cells and CD19+ B cells.

### 4.15. Statistical Analysis

The analyses and visualizations in this study were conducted using R software (version 4.4.1). To determine significant differences between groups, we used either the t-test or the Wilcoxon signed-rank test for paired samples, depending on whether the data followed a normal distribution and whether the design was paired. A *p* < 0.05 was considered statistically significant.

## 5. Conclusions

This study identified three key ERS-DEGs—*GPX1*, *RCN1*, and *UBE2D3*—as biomarkers through bioinformatics analysis and validated them experimentally. *GPX1* and *RCN1* showed consistent and significant expression trends in both datasets and animal models. Joint analysis revealed associations between these genes and immune infiltration. Cluster analysis of MS patients based on these biomarkers not only uncovered new molecular subtypes but also predicted potential drugs targeting different subtypes (such as metformin) and identified differences in immune infiltration among the subtypes. Combined with the previous literature and the results of single-cell sequencing analysis, it is suggested that the expression of GPX1 and RCN1 in microglia and pericytes may regulate endoplasmic reticulum stress through the APP–CD74 pathway, thereby further affecting the progression of MS, while also identifying specific subtypes of microglia and their biological functions. Finally, Mendelian randomization was used to analyze the causal associations between these key ERS-DEGs and MS. This study not only identified MS biomarkers related to endoplasmic reticulum stress but also elucidated their potential mechanisms in regulating MS through specific cells and pathways, providing new research directions and potential drug development targets for MS treatment.

## Figures and Tables

**Figure 1 ijms-26-06286-f001:**
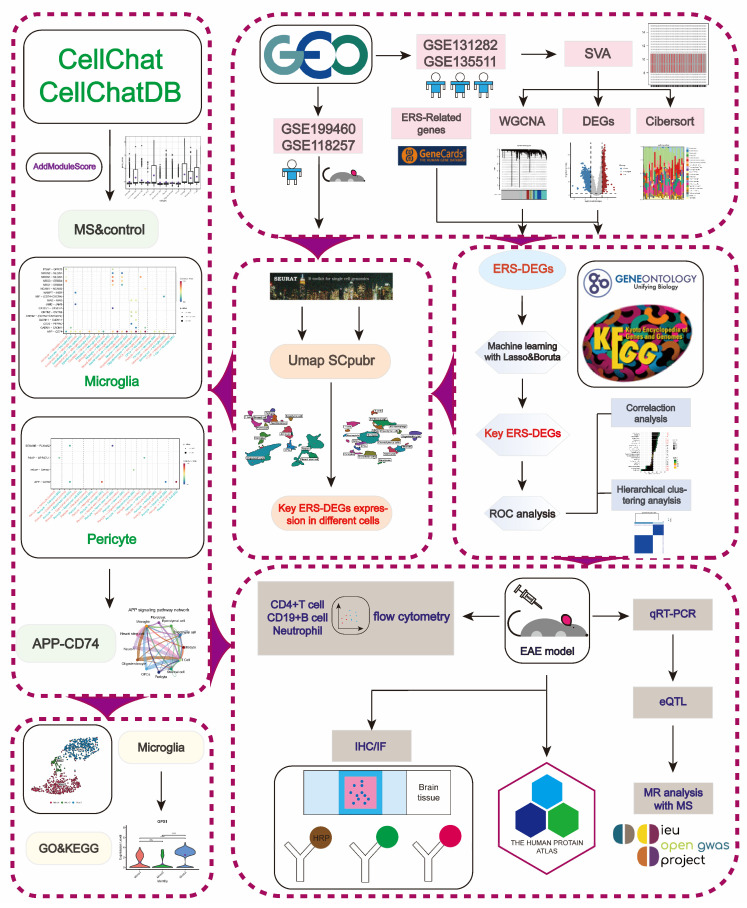
The flowchart of the research work.

**Figure 2 ijms-26-06286-f002:**
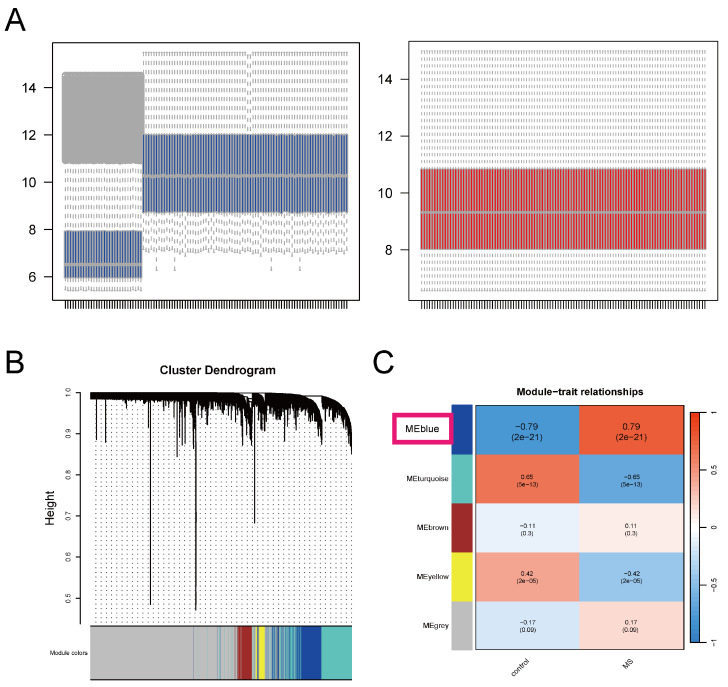
Combining datasets and WGCNA (**A**) Remove batch effects between different datasets. (**B**) The clustering dendrogram of co-expression modules based on topological overlap. (**C**) The association between modules and traits is illustrated by a heatmap, which displays the correlation between genes within the modules and clinical phenotypes. Each row corresponds to a color-coded module, and each column corresponds to a clinical trait. Red indicates positive correlation with the phenotype, while blue indicates negative correlation with the phenotype.

**Figure 3 ijms-26-06286-f003:**
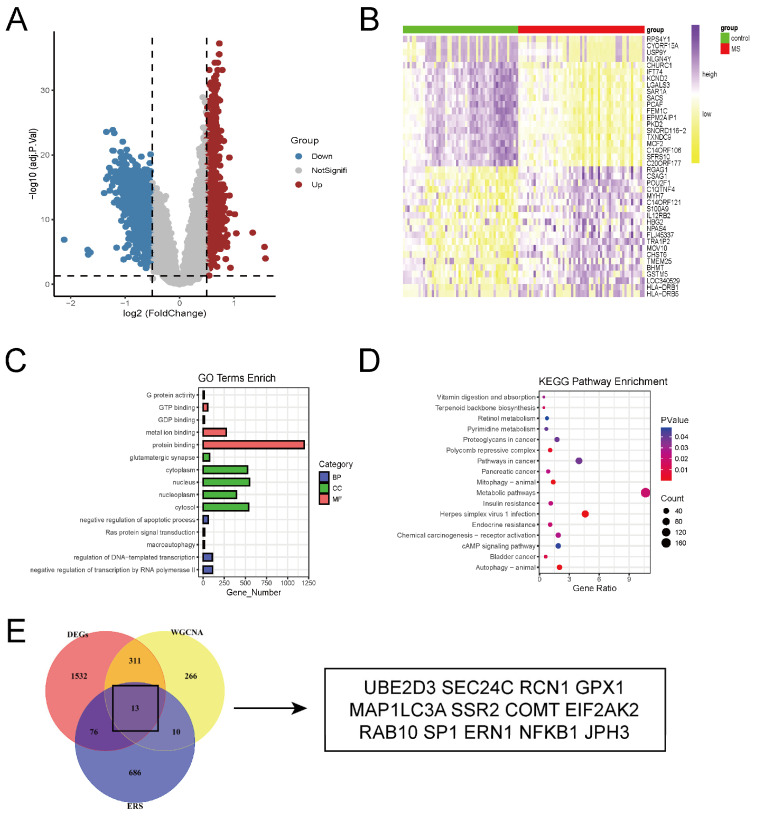
Identification and analysis of DEGs. (**A**) Volcano plot of differentially expressed genes (DEGs). (**B**) Heatmap of top 20 DEGs. (**C**) GO enrichment analysis of DEGs. (**D**) KEGG pathway enrichment analysis of DEGs. (E) The overlapping genes between DEGs, the blue module of WGCNA, and ERS-related genes.

**Figure 4 ijms-26-06286-f004:**
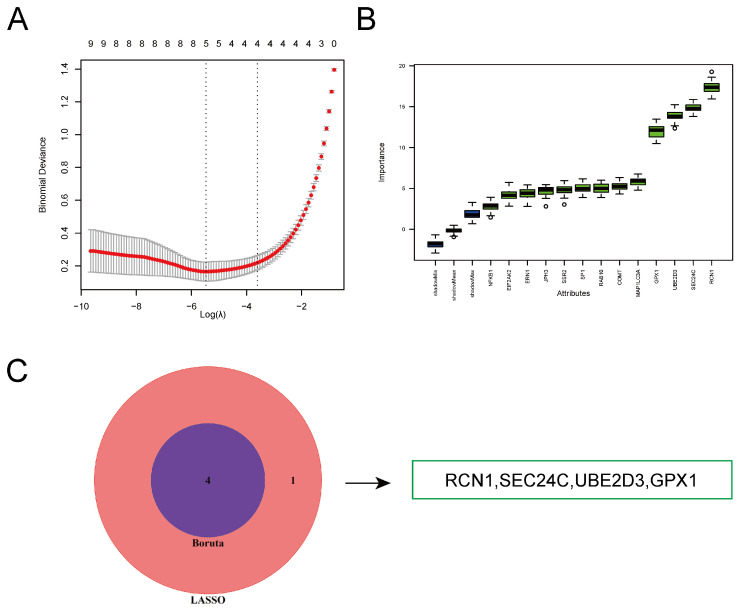
Machine Learning for Identification of Key ERS-DEGs. (**A**) LASSO identified five feature genes. (**B**) The ranking of feature gene importance given by the Boruta algorithm, and obtaining the four feature genes with importance greater than 10. (**C**) The overlapping genes between the feature genes selected by LASSO regression and those selected by Boruta.

**Figure 5 ijms-26-06286-f005:**
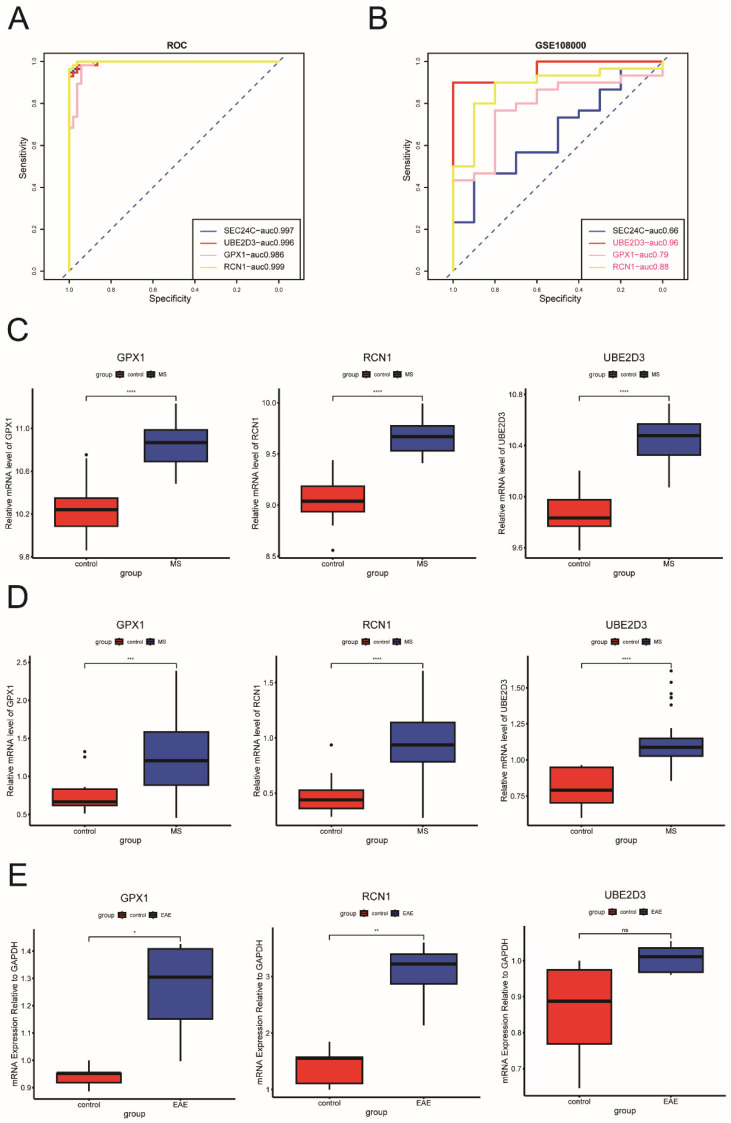
Receiver Operating Characteristic (ROC) analysis and expression profiles of key ERS-DEGs. (**A**) ROC analysis of key ERS-DEGs in the merged dataset. (**B**) ROC analysis of key ERS-DEGs in the external dataset. (**C**) Expression profiles of key ERS-DEGs with an area under the curve (AUC) greater than 0.7 in both datasets, shown in the merged dataset. (**D**) Expression profiles of key ERS-DEGs with an AUC greater than 0.7 in both datasets, shown in the external dataset. (**E**) In vivo validation of key ERS-DEGs with an AUC greater than 0.7 in both datasets in the EAE disease model (*n* = 5) and control group (*n* = 5) (ns: not significant, * Denotes *p* < 0.05, ** *p* < 0.01, *** *p* < 0.001 and **** *p* < 0.0001, *t*-test).

**Figure 6 ijms-26-06286-f006:**
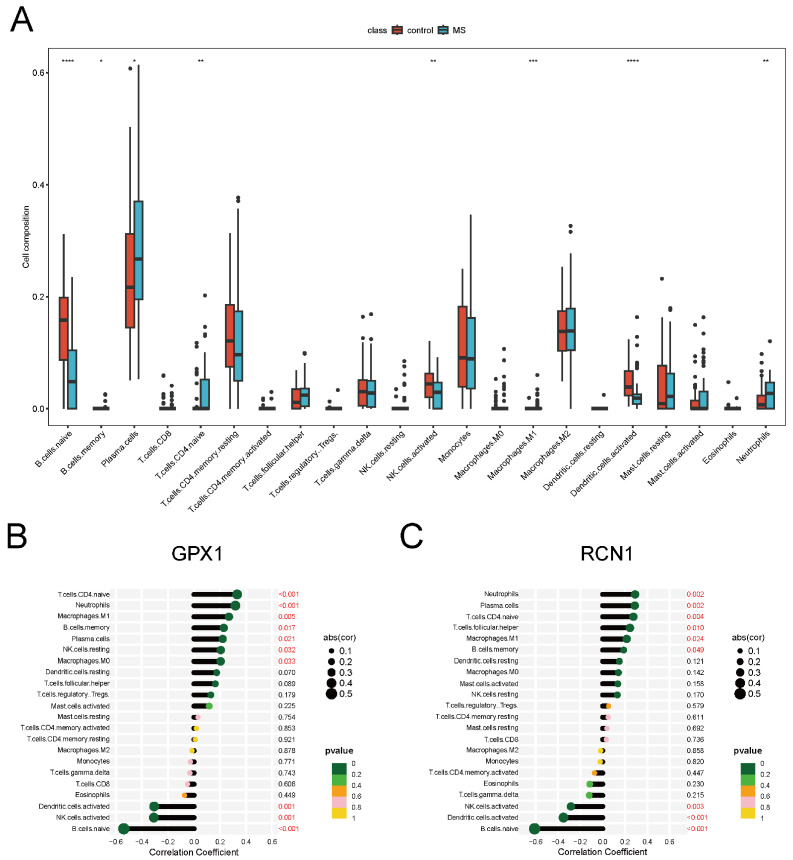
Immune infiltration analysis. (**A**) Levels of immune cell fractions between MS and control samples. (**B**) Correlation of *GPX1* with immune cells. (**C**) Correlation of *RCN1* with immune cells. (* Denotes *p* < 0.05, ** *p* < 0.01, *** *p* < 0.001 and **** *p* < 0.0001, *t*-test).

**Figure 7 ijms-26-06286-f007:**
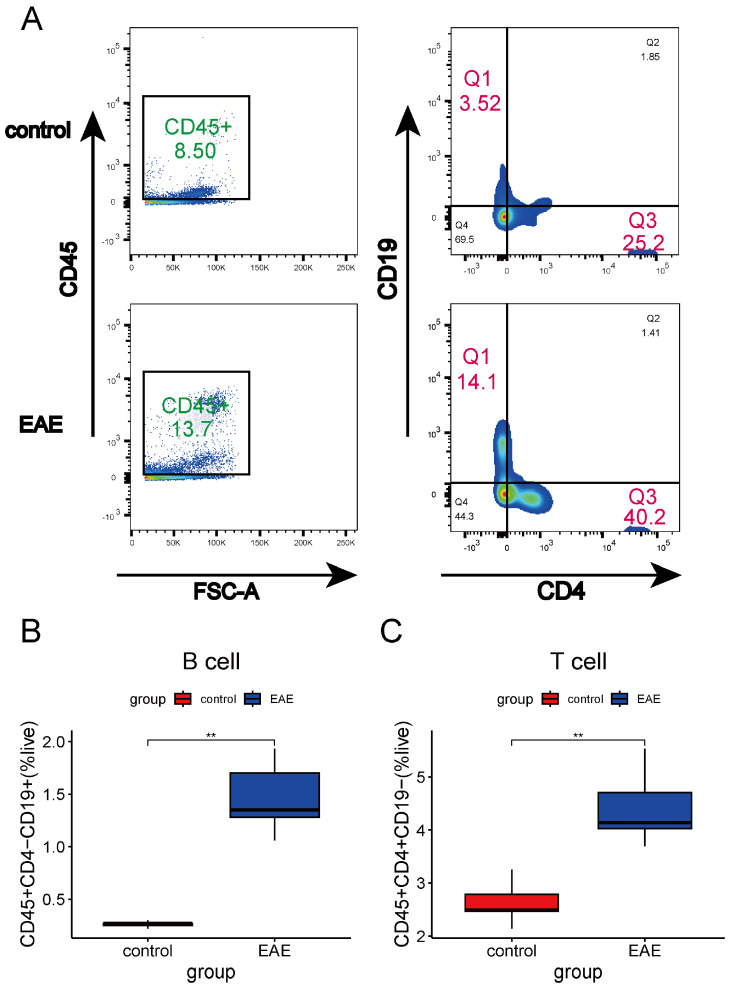
Proportion of T cells and B cells in the brain tissue of EAE mice. (**A**) Flow cytometry analysis of B cells and T cells. (**B**) Comparison of B cell proportions between EAE and the control group. (**C**) Comparison of T cell proportions between EAE and the control group (n = 5) (** *p* < 0.01, *t*-test).

**Figure 8 ijms-26-06286-f008:**
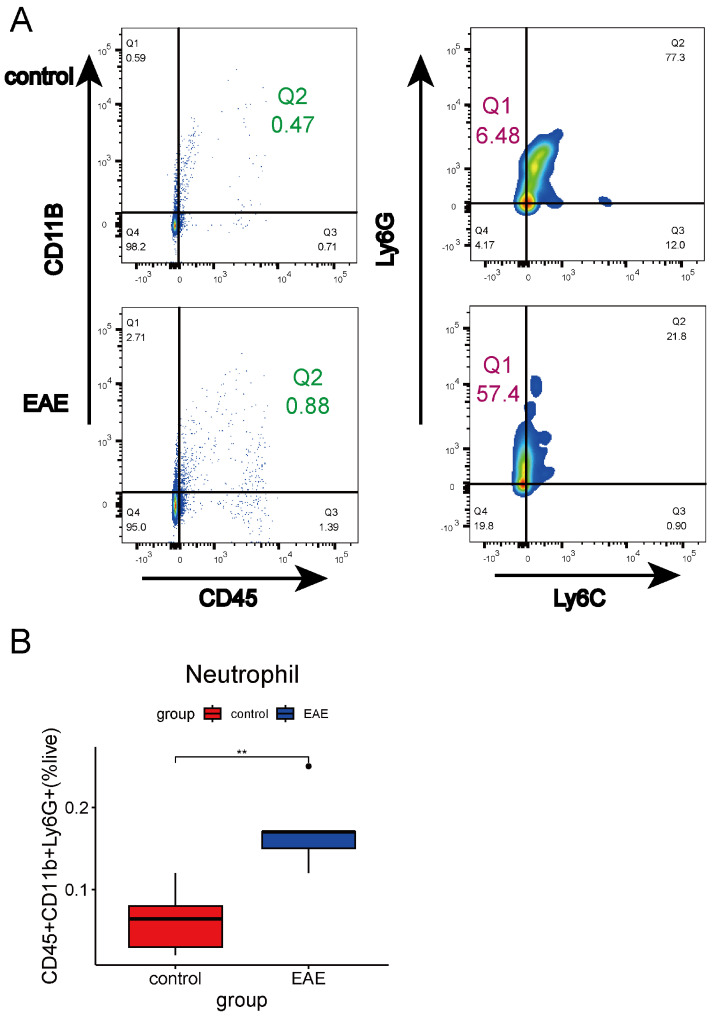
Proportion of neutrophils in the brain tissue of EAE mice. (**A**) Flow cytometry analysis of neutrophils. (**B**) Comparison of neutrophil proportions between EAE and the control group (n = 5) (** *p* < 0.01, *t*-test).

**Figure 9 ijms-26-06286-f009:**
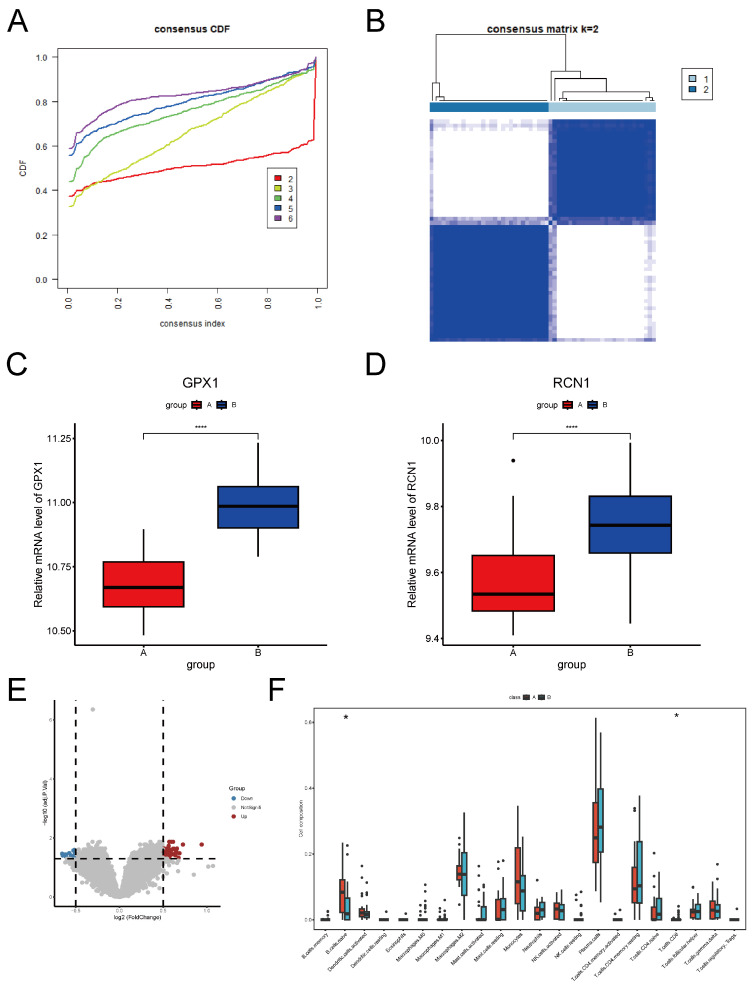
Consensus clustering and analysis of MS. (**A**) Consensus cumulative distribution function plot. (**B**) Heatmap of consensus clustering analysis (k = 2). (**C**,**D**) Expression of *GPX1* and *RCN1* in Group A and Group B. (**E**) Volcano plot of DEGs between Group A and Group B. (**F**) The immune cell composition of Group A and Group B. (* Denotes *p* < 0.05, **** *p* < 0.0001, *t*-test).

**Figure 10 ijms-26-06286-f010:**
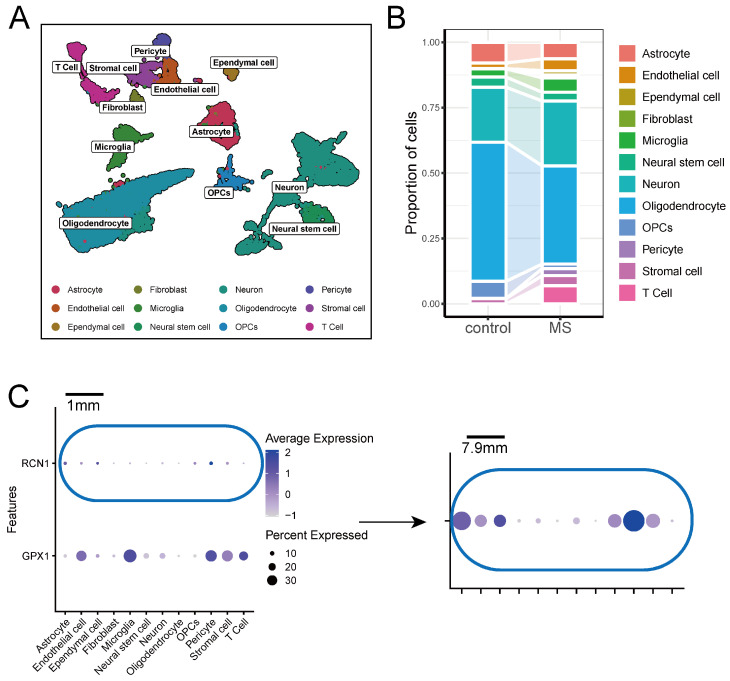
Single-nucleus transcriptomic analysis of key ERS-DEGs. (**A**) UMAP plot of GSE118257. (**B**) Stacked bar chart of cell proportions. (**C**) Expression of *GPX1* and *RCN1* in GSE118257.

**Figure 11 ijms-26-06286-f011:**
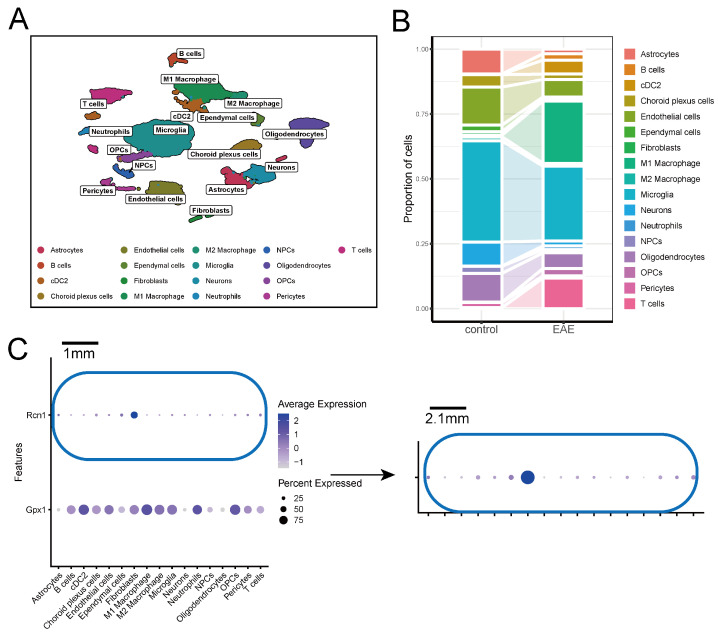
Single-cell transcriptomic analysis of key ERS-DEGs. (**A**) UMAP plot of GSE199460. (**B**) Stacked bar chart of cell proportions. (**C**) Expression of *GPX1* and *RCN1* in GSE199460.

**Figure 12 ijms-26-06286-f012:**
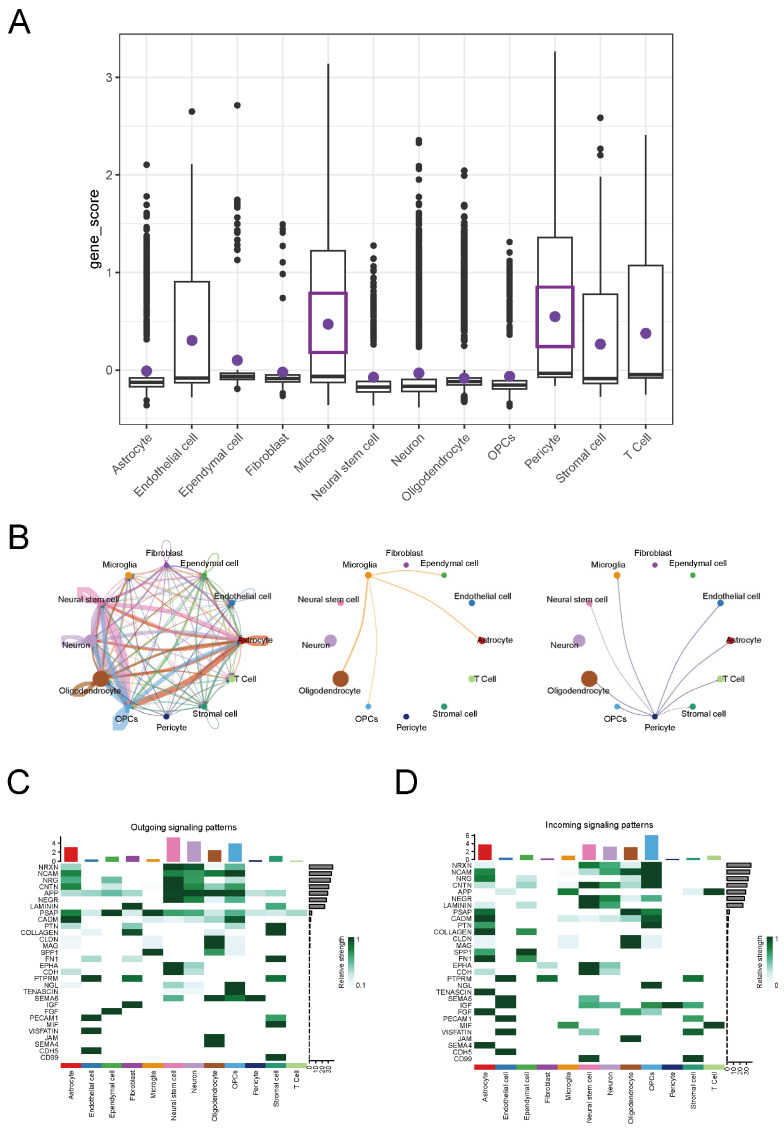
Cell Communication Analysis. (**A**) Boxplot of cell scoring. (**B**) Chord diagram of cell communication among all cells, microglia, and pericytes. (**C**,**D**) Heatmap of signal outgoing and incoming for all pathways.

**Figure 13 ijms-26-06286-f013:**
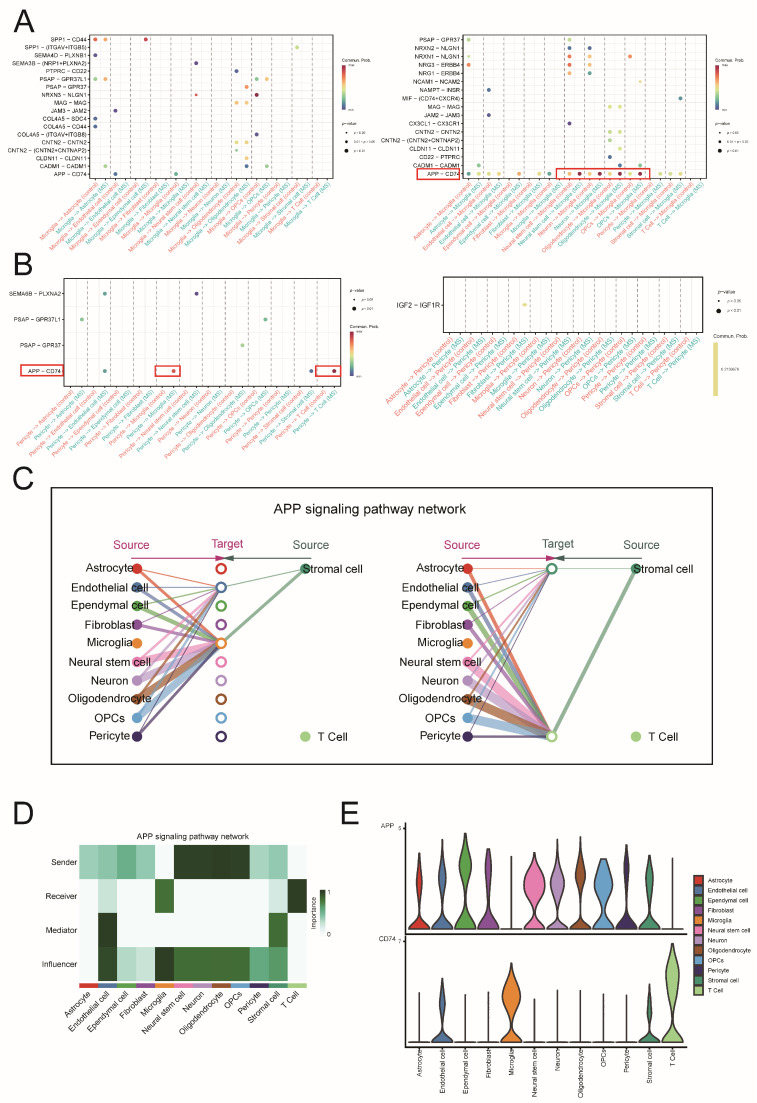
APP-CD74 pathway in MS. (**A**) Dotplot compares the pathways of microglia as a signal sender and receiver. (**B**) Dotplot compares the pathways of pericytes as a signal sender and receiver. (**C**) The hierarchical plot of the APP-CD74 pathway. (**D**) Heatmap displaying the network centrality scores. (**E**) Violin plot of *APP* and *CD74* expression.

**Figure 14 ijms-26-06286-f014:**
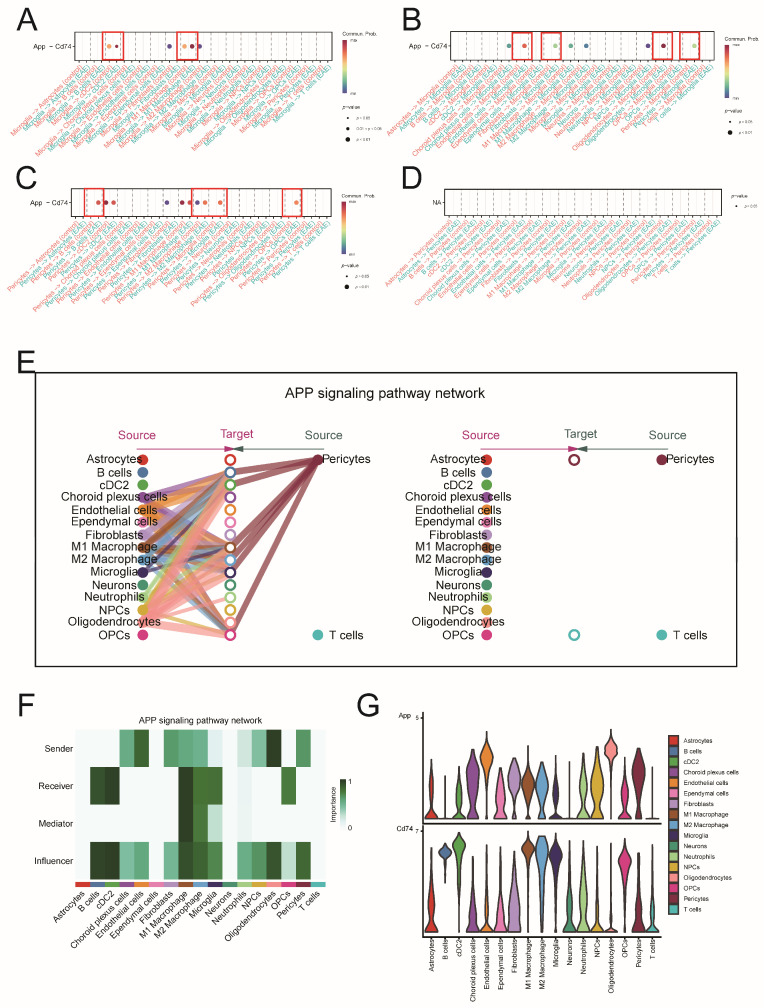
APP-CD74 pathway in EAE. (**A**,**B**) Dotplot compares the APP-CD74 pathway of microglia as signal senders and receivers. (**C**,**D**) Dotplot compares the APP-CD74 pathway of pericytes as signal senders and receivers. (**E**) The hierarchical plot of the APP-CD74 pathway. (**F**) Heatmap displaying the network centrality scores. (**G**) Violin plot of *APP* and *CD74* expression.

**Figure 15 ijms-26-06286-f015:**
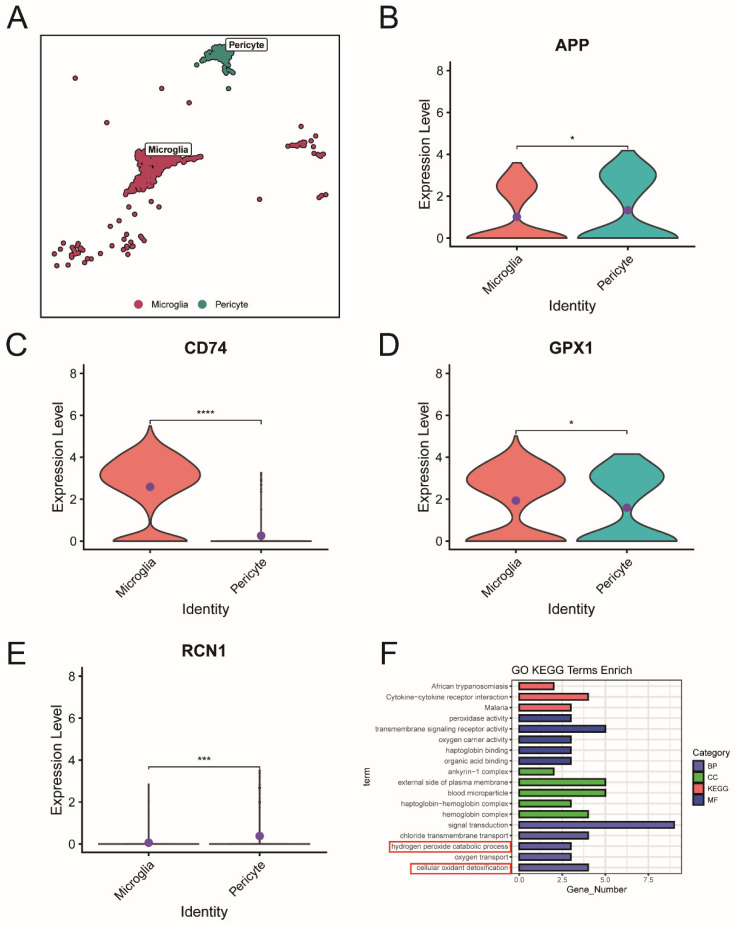
Expression differences of ligand–receptor pair and key ERS-DEGs in microglia and pericytes in MS. (**A**) Pericytes and microglia in MS. (**B**–**E**) Expression differences of *APP*, *CD74*, *GPX1*, and *RCN1* between microglia and pericytes in MS. (**F**) GO and KEGG enrichment analysis of pericytes. (* Denotes *p* < 0.05, *** *p* < 0.001 and **** *p* < 0.0001, *t*-test).

**Figure 16 ijms-26-06286-f016:**
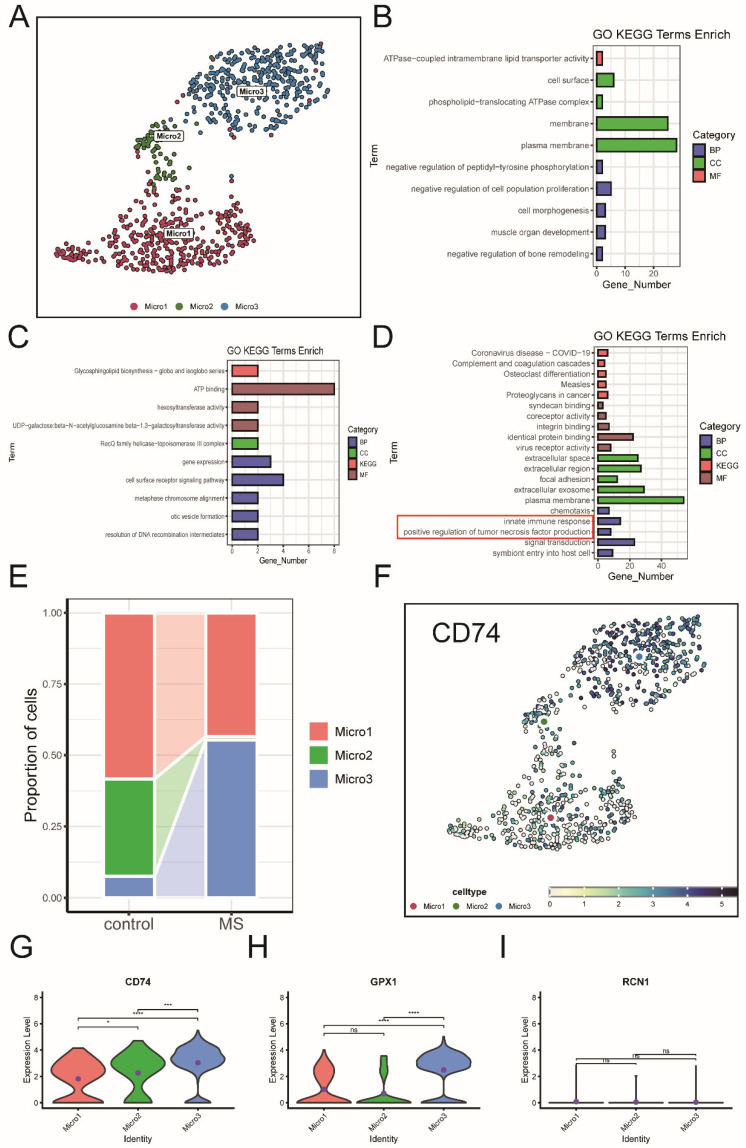
Subtype analysis of microglia. (**A**) Microglia are divided into three subtypes. (**B**–**D**) GO and KEGG enrichment analysis of the three microglia subtypes. (**E**) Stacked bar chart of cell proportions. (**F**) Variations of CD74 among the three microglia subtypes. (**G**–I) Expression differences of *CD74*, *GPX1*, and *RCN1* among the three microglia subtypes. (ns: not significant, * Denotes *p* < 0.05, *** *p* < 0.001 and **** *p* < 0.0001, *t*-test).

**Figure 17 ijms-26-06286-f017:**
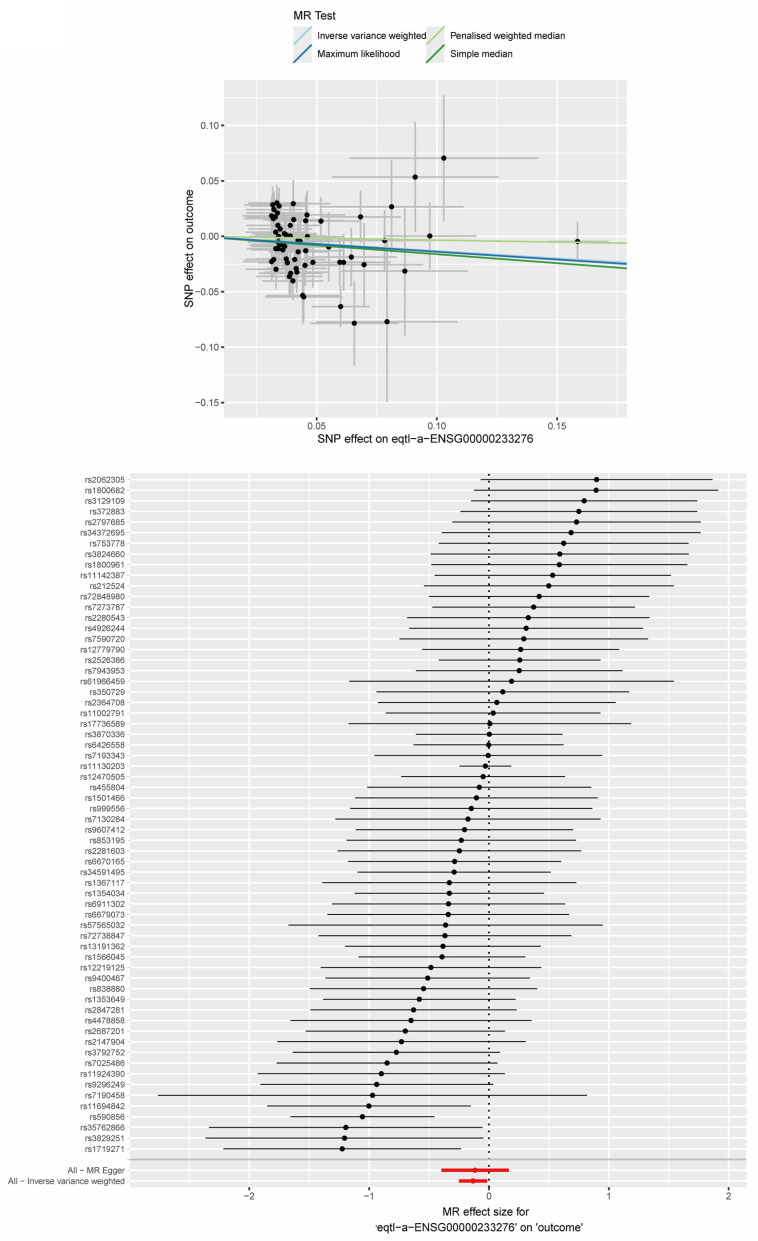
Two-sample Mendelian randomization (MR) analysis for *GPX1* and MS. Scatter plot, forest plot, and leave-one-out sensitivity analysis of the two-sample MR analysis for *GPX1* and MS.

**Figure 18 ijms-26-06286-f018:**
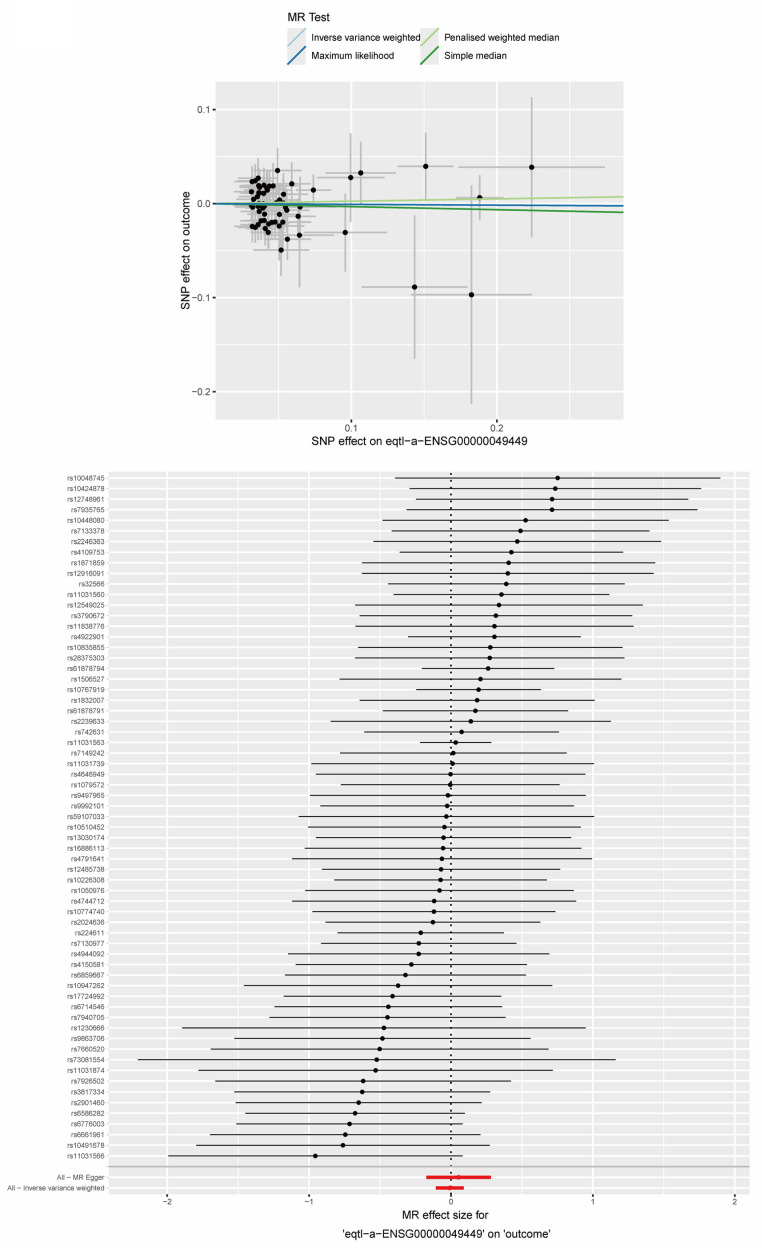
Two-sample Mendelian randomization (MR) analysis for *RCN1* and MS. Scatter plot, forest plot, and leave-one-out sensitivity analysis of the two-sample MR analysis for RCN1 and MS.

## Data Availability

The data supporting the results reported in this article can be provided upon reasonable request.

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
