# Peer review of "GPX1 and RCN1 as New Endoplasmic Reticulum Stress-Related Biomarkers in Multiple Sclerosis Brain Tissue and Their Involvement in the APP-CD74 Pathway: An Integrated Study Combining Machine Learning and Multi-Omics"

_ijms, 2025, doi:10.3390/ijms26136286_

Round 1
Reviewer 1 Report
Comments and Suggestions for Authors
This study integrates WGCNA, differential gene expression, and machine learning to identify GPX1 and RCN1 as key endoplasmic reticulum stress (ERS)-related biomarkers in multiple sclerosis (MS) brain tissue. The authors validate their relevance through external datasets and animal models, demonstrate their association with pro-inflammatory immune cell infiltration, and explore their expression and function at the single-cell level, particularly in microglia and pericytes. They uncover a potential mechanism involving the APP-CD74 signaling pathway, propose molecular subtypes of MS based on ERS gene expression, and identify potential therapeutic compounds. Mendelian randomization further suggests a protective role for GPX1 in MS, providing new insights into MS pathogenesis and candidate targets for treatment.
However, the manuscript has disorganized writing and poorly presented figures, both of which require substantial revision and reorganization. Thus, we recommended major revision.
- The manuscript is written in a stepwise manner according to the methods used, but it lacks clear rationale for each analytical step. Most result interpretations are limited to figure descriptions, without deeper biological insights or contextual integration with existing knowledge. The writing is not standardized, for example, gene names should be italicized, and p-value formatting needs to be consistent throughout. The aim of the study is not clearly stated in the title and abstract, making it difficult to understand the main objective. Analytical methods and techniques are described as the aim, which is inappropriate. The description of the study content is overly lengthy and method-heavy, lacking a clear connection between the background, objective, and key findings. A more concise and focused abstract is recommended, emphasizing the biological question, the main findings, and their significance.There are a total of 16 figures, many of which could be moved to the supplementary materials; several figures also contain text that is difficult to read. The overall storyline is unclear. It is recommended to restructure the Results section into 5–7 focused subsections, each corresponding to a well-organized figure panel.
- Figure 1 and Figure 10 to 16 are not referenced in the main text, making it unclear which results correspond to which figures. All figures should be explicitly cited and integrated into the narrative to guide the reader and ensure clarity.
- There is confusion regarding the data source used for enrichment analysis in Figures 3D and 3E (note: Figure 3E is mislabeled as 3D). Please clarify whether the enrichment analysis was performed on all DEGs or on the intersection of DEGs, WGCNA modules, and ERS-related genes. If it was based solely on DEGs, it may not be appropriate to present this after Figure 3C, which integrates WGCNA and ERS-related information. Additionally, the visual styles of Figures 3D and 3E should be consistent. In Figure 3D, p-value distributions are not shown, which makes it difficult to evaluate the statistical significance of the enriched pathways. The relevance of the enriched pathways to MS should be more thoroughly explained, especially why cancer-related pathways were enriched and how they relate to MS pathogenesis?Furthermore, the manuscript should address why there is only a small overlap between DEGs and ERS-related genes. Does this suggest a weak association between ER stress and MS? Lastly, please justify the rationale for using both DEG and WGCNA approaches for gene selection. Why not rely on a single method, or what does the integration add biologically?
- Are there any well-established or widely accepted biomarkers for multiple sclerosis, such as OLIG2, MOG, or CHIT1, and if so, how do these perform in your dataset? Have these known biomarkers been included as references or benchmarks for comparison? It would strengthen the study to demonstrate whether your identified biomarkers (GPX1 and RCN1) outperform or complement existing ones.
- The rationale for using only two biomarkers (GPX1 and RCN1) to perform molecular subtype classification requires further clarification. Is the use of just two genes sufficient to capture the heterogeneity of MS? Beyond molecular distinctions, are there any phenotypic differences between the identified subtypes, such as clinical outcomes (e.g., survival), sex, age, or disease stage? Moreover, are these two molecular subtypes consistently observed across multiple independent datasets or batches, and does their proportion remain relatively stable?
- Regarding drug target identification, how were the subtype-specific drug targets predicted, and have these targets been validated across different datasets? The study should also address whether these targets are biologically meaningful and relevant to MS pathogenesis. Additionally, the observed differences in immune cell infiltration between subtypes should be interpreted in more depth. What is the biological or clinical implication of these variations in immune composition? Are these differences potentially actionable or do they suggest functional divergence between subtypes?
- There appears to be inconsistency between Figure 10C, Figure 11C, and Figure 6D/6E.The relationship among these figures needs to be clarified, how should these results be interpreted collectively? What is their mechanistic or correlative contribution to the immune landscape in MS? Additionally, Figure 10D and Figure 10E are missing key control groups, such as control T cells and control ependymal cells, which are necessary for proper interpretation of the findings.
- In method section, the tools used should be more detailed. For example, which R package were used for background correction, normalization, probeset filtering, and annotation? Which R packages were used for the visualization of the differential analysis? Also, the Methods section should also be reorganized, with a clear separation between experimental procedures and bioinformatics analyses.
Author Response
Comments 1:The manuscript is written in a stepwise manner according to the methods used, but it lacks clear rationale for each analytical step. Most result interpretations are limited to figure descriptions, without deeper biological insights or contextual integration with existing knowledge. The writing is not standardized, for example, gene names should be italicized, and p-value formatting needs to be consistent throughout. The aim of the study is not clearly stated in the title and abstract, making it difficult to understand the main objective. Analytical methods and techniques are described as the aim, which is inappropriate. The description of the study content is overly lengthy and method-heavy, lacking a clear connection between the background, objective, and key findings. A more concise and focused abstract is recommended, emphasizing the biological question, the main findings, and their significance.There are a total of 16 figures, many of which could be moved to the supplementary materials; several figures also contain text that is difficult to read. The overall storyline is unclear. It is recommended to restructure the Results section into 5–7 focused subsections, each corresponding to a well-organized figure panel.
Response 1:Thank you for taking the time out of your busy schedule to review our manuscript. Your comments have been extremely helpful in further improving our work. Regarding the abstract, we have revised it as a whole, reducing redundancy in length while highlighting the biological questions and key findings. We have also merged some of our results and reorganized certain figures and tables, moving them to the supplementary files.
Comments 2: Figure 1 and Figure 10 to 16 are not referenced in the main text, making it unclear which results correspond to which figures. All figures should be explicitly cited and integrated into the narrative to guide the reader and ensure clarity.
Response 2:Thank you for pointing out the issue. We have now labeled Figures 1 and 10 to 16 in the corresponding sections of the article.
Comments 3: There is confusion regarding the data source used for enrichment analysis in Figures 3D and 3E (note: Figure 3E is mislabeled as 3D). Please clarify whether the enrichment analysis was performed on all DEGs or on the intersection of DEGs, WGCNA modules, and ERS-related genes. If it was based solely on DEGs, it may not be appropriate to present this after Figure 3C, which integrates WGCNA and ERS-related information. Additionally, the visual styles of Figures 3D and 3E should be consistent. In Figure 3D, p-value distributions are not shown, which makes it difficult to evaluate the statistical significance of the enriched pathways. The relevance of the enriched pathways to MS should be more thoroughly explained, especially why cancer-related pathways were enriched and how they relate to MS pathogenesis?Furthermore, the manuscript should address why there is only a small overlap between DEGs and ERS-related genes. Does this suggest a weak association between ER stress and MS? Lastly, please justify the rationale for using both DEG and WGCNA approaches for gene selection. Why not rely on a single method, or what does the integration add biologically?
Response 3:Thank you for pointing out the deficiencies in our interpretation of the results. The enrichment analysis was conducted on all differentially expressed genes (DEGs). Therefore, we have reorganized the images in Figure 3. After performing the enrichment analysis on Figure 3D and Figure 3E, we selected the enrichment results with p < 0.05. Given the large number of Gene Ontology (GO) enrichment results, we highlighted the top five most significant enrichment results for each of the Biological Process (BP), Cellular Component (CC), and Molecular Function (MF) categories. Consequently, we have added additional details about the enrichment analysis in the Methods section.
The method of enrichment analysis essentially involves dissecting and summarizing specific gene sets, such as DEGs or other uniquely identified gene collections, within a database of functionally annotated gene sets/pathway gene sets. However, these functions/pathways do not exist in isolation. Particularly for KEGG pathway information, most genes are often present in multiple pathways, and their roles can vary across different pathways. Cancer, as one of the most complex diseases at present, encompasses a vast number of genes in the KEGG database. Therefore, when DEGs have diverse and complex functions, it is easy for them to be enriched in cancer-related KEGG pathways. In such cases, it is necessary to make comprehensive judgments based on all functional and pathway information.
In our Venn diagram, we set three limiting conditions: DEGs-WGCNA module-ERS. When we intersect DEGs with ERS-related genes alone, there are 89 common genes. However, after introducing the WGCNA module, most strongly associated with the trait, the final intersection of DEGs-WGCNA-ERS yields only 13 common genes. This is a widely used method for dimensionality reduction and screening of gene sets, which has been extensively applied in many studies [1]. This is the primary reason for the reduction of our gene set. Additionally, multiple studies have already highlighted the association between ERS and MS and emphasized the need to identify more relevant biomarkers, which is also the purpose of our research. The combination of DEGs and WGCNA for dimensionality reduction is described in detail in this reference [2]. The key advantage of this method is that it allows the introduction of traits (such as the binary trait we introduced in our manuscript) during the exploration of biomarkers, which can facilitate the discovery of biomarkers. Therefore, we chose this approach.
Comments 4: Are there any well-established or widely accepted biomarkers for multiple sclerosis, such as OLIG2, MOG, or CHIT1, and if so, how do these perform in your dataset? Have these known biomarkers been included as references or benchmarks for comparison? It would strengthen the study to demonstrate whether your identified biomarkers (GPX1 and RCN1) outperform or complement existing ones.
Response 4:Thank you for your valuable comments. Since our dataset is an integrated dataset, we compared two well-established multiple sclerosis biomarkers, MOG and OLIG2. Their AUC values in our dataset were 0.808 and 0.777, respectively, both of which are greater than 0.7 (Figure 1), which also demonstrates the stability of these two biomarkers. Our method for screening biomarkers is based on a combined machine learning model. This method selects biomarkers based on fitting the optimal model, and we included external datasets to calculate their AUC values. Only biomarkers with AUC values greater than 0.7 in both datasets and validated in animal models were ultimately retained. This approach has been applied in multiple studies [3]. The final biomarkers we obtained had AUC values exceeding 0.85 in the original dataset, which also proves the feasibility of our screening method.
Figure 1: ROC analysis of key ERS-DEGs, MOG, and OLIG2 in the merged dataset.
Comments 5: The rationale for using only two biomarkers (GPX1 and RCN1) to perform molecular subtype classification requires further clarification. Is the use of just two genes sufficient to capture the heterogeneity of MS? Beyond molecular distinctions, are there any phenotypic differences between the identified subtypes, such as clinical outcomes (e.g., survival), sex, age, or disease stage? Moreover, are these two molecular subtypes consistently observed across multiple independent datasets or batches, and does their proportion remain relatively stable?
Response 5: Thank you for your professional and rigorous suggestions, which will greatly help us accurately elaborate our analytical results. The use of genes screened by machine learning or protein–protein interaction (PPI) networks to subtype disease samples has been widely employed in numerous studies [4, 5], primarily in the field of cancer research. When subtyping tumors based on the survival data of a single gene and comparing these subtypes, differences in gene expression were found between subtypes [6]. However, we fully agree with the issue you pointed out: some limitations exist in distinguishing disease subtypes based solely on two key genes. Therefore, we stated this limitation in the discussion. Since brain tissue samples from multiple sclerosis patients are mostly derived from frozen tissue banks and the corresponding clinical information is relatively limited, the clinical information of the original dataset only includes a part of the gender information, which is also a limitation in the process of subtyping many non-tumor diseases. Therefore, the final subtypes we obtained focus on molecular differences. The two molecular subtypes originated from a merged dataset of two datasets, which had undergone batch correction before subtyping. Due to the limited number of multiple sclerosis brain tissue datasets and the relatively small sample size of most disease samples, whether the molecular subtypes we obtained through analysis apply to other datasets needs to be further analyzed after obtaining more clinical samples in the future. Since the main focus of our study is on the application of machine learning in screening disease biomarkers and the application of these biomarkers, we stated these limitations in the discussion section and pointed out the necessity of further expanding data research and clinical research in the future.
Your suggestions are highly instructive to us. In the future, we will endeavor to collect more clinical samples and obtain as much clinical information as possible, and further deepen our research in the next phase.
Comments 6: Regarding drug target identification, how were the subtype-specific drug targets predicted, and have these targets been validated across different datasets? The study should also address whether these targets are biologically meaningful and relevant to MS pathogenesis. Additionally, the observed differences in immune cell infiltration between subtypes should be interpreted in more depth. What is the biological or clinical implication of these variations in immune composition? Are these differences potentially actionable or do they suggest functional divergence between subtypes?
Response 6:Thank you for your valuable suggestions on improving drug prediction based on subtypes. We used the differential genes between the two subtypes as targets to predict drugs. For the predicted drugs, we only retained those targeting multiple genes and that are “approved.” Due to the limitations of subtyping based solely on bioinformatics, the targets we obtained are temporarily only from our merged dataset. In the future, following your advice, we will use more complex subtyping methods and include more drug experiments and clinical cases in more datasets to further study disease subtypes and specific drug targets. We ultimately retained only two drugs. One of them is metformin, which has been listed as a potential therapeutic drug for multiple sclerosis in relevant literature . Given its numerous drug targets, the effectiveness of these targets needs to be validated through network pharmacology and multi-phase drug experiments. Our dataset is a merged one, incorporating a larger number of clinical samples. Due to the limited availability of brain tissue sequencing samples from multiple sclerosis patients, we utilized the existing samples to the fullest extent possible. The differences in immune cell infiltration between subtypes are actually related to the correlation between biomarkers and immune cell infiltration. When a biomarker is positively correlated with the infiltration of a specific immune cell, this correlation often manifests as differences in immune cell infiltration between subtypes. This difference in immune cell infiltration provides another perspective for evaluating biomarker-related immune cells, and these differences need to be further analyzed through more clinical cases. We stated the potential connections between them and the limitations of the potential operability of the differences in the discussion section.
Your suggestions are of great guiding significance to our future work. In the future, we will incorporate more clinical samples and conduct more complex drug experiments based on the predicted drugs to validate the differences in immune cells and further investigate the relevant targets.
Comments 7: There appears to be inconsistency between Figure 10C, Figure 11C, and Figure 6D/6E.The relationship among these figures needs to be clarified, how should these results be interpreted collectively? What is their mechanistic or correlative contribution to the immune landscape in MS? Additionally, Figure 10D and Figure 10E are missing key control groups, such as control T cells and control ependymal cells, which are necessary for proper interpretation of the findings.
Response 7:Thank you for your important suggestions on the overall interpretation of the figures. Figures 6D and 6E depict the correlation between the biomarkers we identified from the merged dataset and immune cell infiltration as calculated by CIBERSORT. Figures 10C and 11C show the expression levels of our identified biomarkers across different cell types, calculated from snRNA-seq and scRNA-seq data, respectively. These figures are derived from different types of sequencing data and involve distinct computational methods. Their contribution to the immune landscape lies in expanding cellular diversity, encompassing not only the differences in immune cell infiltration from CIBERSORT but also uncovering additional cell types from single-cell data, thereby enhancing the interpretation of the biomarkers. This approach has been widely applied in many studies [7, 8].
The single-cell data used to create Figures 10D and 10E are from a publicly available dataset [9]. In this dataset, there is a lack of immune cells and some other cell types in the control group. Due to the limited availability of public datasets for multiple sclerosis brain tissue, and considering that snRNA-seq of post-mortem brain tissue generally has lower immune cell abundance compared to scRNA-seq from living samples, we supplemented our analysis with an additional scRNA-seq dataset that includes all cell types. This enhances the completeness of the single-cell analysis and improves the interpretation of the results. To maintain clarity, we have placed Figures 10D, 10E, 11D, and 11E together in Supplementary Figure 2.
In the future, we plan to obtain more single-cell sequencing data for multiple sclerosis to conduct more complex follow-up research.
Comments 8: In method section, the tools used should be more detailed. For example, which R package were used for background correction, normalization, probeset filtering, and annotation? Which R packages were used for the visualization of the differential analysis? Also, the Methods section should also be reorganized, with a clear separation between experimental procedures and bioinformatics analyses.
Response 8:Thank you for your suggestions for revision. We have reorganized the Methods section to include more detailed descriptions of the tools used and to distinguish between the experimental procedures and bioinformatics analyses.
Thank you once again for your patience and professionalism, as well as your recognition of our work. Your suggestions are of great guiding significance to our future research. We look forward to your reply.
[1] T. Gong, Y. Liu, Z. Tian, M. Zhang, H. Gao, Z. Peng, S. Yin, C.W. Cheung, Y. Liu, Identification of immune-related endoplasmic reticulum stress genes in sepsis using bioinformatics and machine learning, Front Immunol 13 (2022) 995974.
[2] N. Sánchez-Baizán, L. Ribas, F. Piferrer, Improved biomarker discovery through a plot twist in transcriptomic data analysis, BMC Biol 20(1) (2022) 208.
[3] S. Zhang, Y. Ma, X. Luo, H. Xiao, R. Cheng, A. Jiang, X. Qin, Integrated Analysis of Immune Infiltration and Hub Pyroptosis-Related Genes for Multiple Sclerosis, J Inflamm Res 16 (2023) 4043-4059.
[4] Y. Ma, F. Wang, Q. Zhao, L. Zhang, S. Chen, S. Wang, Identifying Diagnostic Markers and Constructing Predictive Models for Oxidative Stress in Multiple Sclerosis, Int J Mol Sci 25(14) (2024).
[5] X. Song, Z. Wang, Z. Tian, M. Wu, Y. Zhou, J. Zhang, Identification of Key Ferroptosis-Related Genes in the Peripheral Blood of Patients with Relapsing-Remitting Multiple Sclerosis and Its Diagnostic Value, Int J Mol Sci 24(7) (2023).
[6] Z. Zhao, C. Sun, J. Hou, P. Yu, Y. Wei, R. Bai, P. Yang, Identification of STEAP3-based molecular subtype and risk model in ovarian cancer, J Ovarian Res 16(1) (2023) 126.
[7] N. Shan, Y. Shang, Y. He, Z. Wen, S. Ning, H. Chen, Common biomarkers of idiopathic pulmonary fibrosis and systemic sclerosis based on WGCNA and machine learning, Sci Rep 15(1) (2025) 610.
[8] X. Cai, J. Deng, X. Zhou, K. Wang, H. Cai, Y. Yan, J. Jiang, J. Yang, J. Gu, Y. Zhang, Y. Ding, Q. Sun, W. Wang, Comprehensive analysis of cuproptosis-related genes involved in immune infiltration and their use in the diagnosis of hepatic ischemia-reperfusion injury: an experimental study, Int J Surg 111(1) (2025) 242-256.
[9] S. Jäkel, E. Agirre, A. Mendanha Falcão, D. van Bruggen, K.W. Lee, I. Knuesel, D. Malhotra, C. Ffrench-Constant, A. Williams, G. Castelo-Branco, Altered human oligodendrocyte heterogeneity in multiple sclerosis, Nature 566(7745) (2019) 543-547.

Reviewer 2 Report
Comments and Suggestions for Authors
- Figure Clarity:
Several figures are not clear enough. It is essential to update these figures to ensure that all details are visible and legible. High-resolution images and clear labeling will enhance the reader's ability to understand the results. - Relevance of Figures to Text:
The manuscript contains a large number of figures, some of which do not directly support the conclusions. It is recommended to simplify the paper by removing less important figures and placing them in the supplementary materials. This will help readers focus on the key findings and improve the paper's conciseness. - Software Parameters:
Provide detailed descriptions of the parameters used for the software and tools employed in the analysis. This will enable other researchers to replicate the study and verify the results. Include specific settings and criteria used in each step of the analysis. - Overall Simplification:
Consider streamlining the manuscript by focusing on the most critical figures and results. This will enhance the clarity and impact of the research.
The English language quality of the paper is relatively good, with clear and accurate expression of research content. However, simplify some redundant expressions to enhance conciseness and clarity, helping readers better understand the core content of the paper.
Author Response
Comments 1: Figure Clarity:
Several figures are not clear enough. It is essential to update these figures to ensure that all details are visible and legible. High-resolution images and clear labeling will enhance the reader's ability to understand the results.
Response 1: Thank you for taking the time out of your busy schedule to review our manuscript. Your suggestions are of great value to the improvement of our draft. We have processed Figures 7 and 8, as well as Figures 10 to 14, to make them clearer.
Comments 2: Relevance of Figures to Text:
The manuscript contains a large number of figures, some of which do not directly support the conclusions. It is recommended to simplify the paper by removing less important figures and placing them in the supplementary materials. This will help readers focus on the key findings and improve the paper's conciseness.
Response 2: Thank you for pointing out the problem of redundancy in our figures. We have removed B-D in Figure 9 and moved B and C in Figure 6, D and E in Figure 10, D and E in Figure 11, B and C in Figure 12, C and E-G in Figure 13, E-H and J in Figure 14, as well as Figure 16, to the supplementary files.
Comments 3: Software Parameters:
Provide detailed descriptions of the parameters used for the software and tools employed in the analysis. This will enable other researchers to replicate the study and verify the results. Include specific settings and criteria used in each step of the analysis.
Response 3: We have refined the description of the software parameters. On page 29, line 468, we introduced the “ggplot2” R package used for plotting. On page 30, line 497, we added the criteria for screening drugs from the DGIdb database. On page 30, line 507, we specified the logfc threshold used for cell clustering. On page 30, line 517, we detailed the specific functions used for cell communication analysis and the pathway filtering threshold (min.cell = 10). We appreciate your suggestions for improving the methodology section.
Comments 4: Overall Simplification:
Consider streamlining the manuscript by focusing on the most critical figures and results. This will enhance the clarity and impact of the research.
Response 4: We have revised the overall abstract, reducing redundancy while highlighting the biological questions and main findings. In the Results section, we have merged “Identification of key ERS-DEGs” and “Validation of key ERS-DEGs” into “Identification and Validation of key ERS-DEGs.” We have also combined “Conducting a separate analysis of cell communication via the APP-CD74 pathway in MS” and “Analysis of cellular communication via the APP-CD74 pathway in the EAE disease model” into “Conducting separate analyses of cell communication via the APP-CD74 pathway in MS and the EAE mouse model.” Additionally, we have merged “Key ERS-DEGs and MS GWAS Data Presentation” and “Key ERS-DEGs and MS MR Analysis” into “MR analysis between Key ERS-DEGs and MS.” We appreciate your suggestions for streamlining our manuscript.
Thank you once again for your patience and professionalism, as well as your recognition of our work. Your suggestions are of great guiding significance to our future research. We look forward to your reply.

Reviewer 3 Report
Comments and Suggestions for Authors
The authors have undertaken an extensive series of analyses with remarkable dedication, and it is readily apparent that the execution of this study was highly demanding. The concept of this manuscript is very interesting. However, there are several questions which the authors should address. Some of questions are listed below.
- In discussion, the authors described that “We found that the APP-CD74 pathway between microglia and pericytes is significantly enhanced in MS. Moreover, by analyzing another scRNA-seq dataset, we discovered that this enhancement also occurs in the brain tissue of EAE mice. Based on the above results and the immune phenotypes of microglia in MS [36], we further conducted a subpopulation analysis of microglia. In the APP-CD74 pathway, microglia, as important receptors, express CD74. We found that in the subpopulation of microglia with higher CD74 expression, GPX1 is also highly expressed. Because the expression of NF-κB during inflammation can increase GPX1 expression, and in MS, the expression of NF-κB does indeed increase, and the deposition of β-amyloid can exacerbate endoplasmic reticulum stress. However, the activation of the APP-CD74 pathway can reduce the deposition of β-amyloid [37-40]. Therefore, GPX1 may function by affecting the ability of microglia to handle β-amyloid through the APP-CD74 pathway, thereby further alleviating endoplasmic reticulum stress. As one of the signal senders of the APP-CD74 pathway in MS, although pericytes are a relatively small cell population, they play an important role in maintaining the blood–brain barrier [41].” In fact, the author’s results from the violin plot indicate that in MS, microglia primarily express CD74, while pericytes mainly express APP. However, there are some questions. What is the function of APP in pericytes? Is its expression induced only under ER stress conditions? It remains unclear whether APP is secreted in its full-length form or as Aβ. Moreover, it is still unknown whether CD74 functionally acts as a receptor for APP. At the very least, immunohistochemistry should be used to show the localization and the expressing cells of both CD74 and APP.
- The pathological consequences of the interaction between pericytes and microglia in MS and EAE remain poorly understood. If there is evidence demonstrating direct interaction between pericytes and microglia in tissue, it should be provided.
Author Response
Comments 1: In discussion, the authors described that “We found that the APP-CD74 pathway between microglia and pericytes is significantly enhanced in MS. Moreover, by analyzing another scRNA-seq dataset, we discovered that this enhancement also occurs in the brain tissue of EAE mice. Based on the above results and the immune phenotypes of microglia in MS [36], we further conducted a subpopulation analysis of microglia. In the APP-CD74 pathway, microglia, as important receptors, express CD74. We found that in the subpopulation of microglia with higher CD74 expression, GPX1 is also highly expressed. Because the expression of NF-κB during inflammation can increase GPX1 expression, and in MS, the expression of NF-κB does indeed increase, and the deposition of β-amyloid can exacerbate endoplasmic reticulum stress. However, the activation of the APP-CD74 pathway can reduce the deposition of β-amyloid [37-40]. Therefore, GPX1 may function by affecting the ability of microglia to handle β-amyloid through the APP-CD74 pathway, thereby further alleviating endoplasmic reticulum stress. As one of the signal senders of the APP-CD74 pathway in MS, although pericytes are a relatively small cell population, they play an important role in maintaining the blood–brain barrier [41].” In fact, the author’s results from the violin plot indicate that in MS, microglia primarily express CD74, while pericytes mainly express APP. However, there are some questions. What is the function of APP in pericytes? Is its expression induced only under ER stress conditions? It remains unclear whether APP is secreted in its full-length form or as Aβ. Moreover, it is still unknown whether CD74 functionally acts as a receptor for APP. At the very least, immunohistochemistry should be used to show the localization and the expressing cells of both CD74 and APP.
Response 1: Thank you for taking the time out of your busy schedule to review our manuscript. Your constructive suggestions are very helpful for us to further improve our conclusions. We agree that under pathological conditions such as MS and EAE, the direct interaction between pericytes and microglia in the CNS is still an unresolved yet important issue. In proliferative diabetic retinopathy (PDR), pericytes communicate globally through the APP signaling pathway, and APP-CD74 is the main ligand-receptor pair. Microglia are one of the important recipient cells of the APP-CD74 signaling pathway [1]. The enzymatic cleavage of APP produces Aβ. In the brain vascular system of the CNS, APP and Aβ are in a dynamic balance. In Alzheimer’s disease (AD), co-localization of Aβ with pericytes has been observed [2–4]. This suggests that pericytes may be involved in the dynamic balance between APP and Aβ. We agree with your question that according to current research, Aβ and APP have been detected in MS and Aβ has potential predictive value for the progression of MS disease, so we believe that Aβ is an important protein for APP gene expression to exert its function [5]. In addition, there are studies showing that the process of APP producing Aβ through splicing is a selective splicing (AS) process, which is a potential target for epigenetic therapeutic intervention [6, 7]. However, our current research based on machine learning and multi-omics cannot determine the specific proportion relationship, and more complex instruments and more precise experiments are needed to explore this splicing process. Your suggestion is the direction for our future further exploration. APP is not only induced under endoplasmic reticulum stress conditions, but also gene mutations, inflammation, oxidative stress, and enhanced neuronal activity can lead to upregulation of APP expression in the CNS. This is a rather complex mechanism. Studies have confirmed the interaction between APP and CD74 [8], and we have cited this study in the discussion section. Aβ, as a metabolic product of APP enzymatic cleavage, is one of the causes of AD pathogenesis and also affects neuroinflammation. In the APP-CD74 pathway, pericytes may mainly reduce Aβ through APP, which may be a potential mechanism for the dynamic balance between APP and Aβ.
Given that our APP-CD74 pathway information comes from human samples in public databases and brain tissue pathological sections of multiple sclerosis (MS) are difficult to obtain, we searched the HPA database for the immunohistochemical results of APP in normal and inflamed brain tissue sections. In the figure, we marked the brown staining of brain vascular layer cells in brain tissue sections under inflammatory conditions, which is consistent with the conclusions we obtained from relevant literature, that is, APP is widely expressed in the neurovascular unit, and pericytes are indeed an important component of the neurovascular unit. CD74 is one of the markers of activated microglia, which has been confirmed in MS [9]. However, we fully acknowledge that these IHC data are not from MS patients or our experimental tissues and should only be regarded as indirect supportive evidence. Therefore, we have made corresponding statements on some limitations of the study in the discussion section. The immunohistochemical images are included in Supplementary Figure 5 of the manuscript.
We fully agree with the reviewer’s opinion that in situ evidence, such as co-localization through immunofluorescence or spatial transcriptomics, would be the most ideal way to support direct cell-to-cell interaction. However, the difficulty in obtaining MS brain tissue has led us to try to obtain indirect supportive evidence from public immunohistochemical resources as much as possible. We have clearly pointed out this limitation in the revised manuscript and emphasized the necessity of using clinical pathological tissues and additional imaging methods for experimental verification in the future. Your opinions are of great guiding value to our future research.
Comments 2: The pathological consequences of the interaction between pericytes and microglia in MS and EAE remain poorly understood. If there is evidence demonstrating direct interaction between pericytes and microglia in tissue, it should be provided.
Response 2: In diabetic retinopathy (DR), microglia activated by STAT3 induce apoptosis of pericytes through signaling[10]. This finding confirms the existence of interactions between microglia and pericytes.
Thank you once again for your patience and professionalism, as well as your recognition of our work. Your suggestions are of great guiding significance to our future research. We look forward to your reply.
[1] X. Xu, C. Zhang, G. Tang, N. Wang, Y. Feng, Single-cell transcriptome profiling highlights the role of APP in blood vessels in assessing the risk of patients with proliferative diabetic retinopathy developing Alzheimer's disease, Front Cell Dev Biol 11 (2023) 1328979.
[2] J. Li, M. Li, Y. Ge, J. Chen, J. Ma, C. Wang, M. Sun, L. Wang, S. Yao, C. Yao, β-amyloid protein induces mitophagy-dependent ferroptosis through the CD36/PINK/PARKIN pathway leading to blood-brain barrier destruction in Alzheimer's disease, Cell Biosci 12(1) (2022) 69.
[3] S. De Schepper, J.Z. Ge, G. Crowley, L.S.S. Ferreira, D. Garceau, C.E. Toomey, D. Sokolova, J. Rueda-Carrasco, S.H. Shin, J.S. Kim, T. Childs, T. Lashley, J.J. Burden, M. Sasner, C. Sala Frigerio, S. Jung, S. Hong, Perivascular cells induce microglial phagocytic states and synaptic engulfment via SPP1 in mouse models of Alzheimer's disease, Nat Neurosci 26(3) (2023) 406-415.
[4] Y.Y. Li, D.D. Guo, R.N. Duan, Y. Li, Interactions between Beta-Amyloid and Pericytes in Alzheimer's Disease, Front Biosci (Landmark Ed) 29(4) (2024) 136.
[5] A.M. Pietroboni, M. Caprioli, T. Carandini, M. Scarioni, L. Ghezzi, A. Arighi, S. Cioffi, C. Cinnante, C. Fenoglio, E. Oldoni, M.A. De Riz, P. Basilico, G.G. Fumagalli, A. Colombi, G. Giulietti, L. Serra, F. Triulzi, M. Bozzali, E. Scarpini, D. Galimberti, CSF β-amyloid predicts prognosis in patients with multiple sclerosis, Mult Scler 25(9) (2019) 1223-1231.
[6] K.V. Nguyen, β-Amyloid precursor protein (APP) and the human diseases, AIMS Neurosci 6(4) (2019) 273-281.
[7] X. Wang, X. Zhou, G. Li, Y. Zhang, Y. Wu, W. Song, Modifications and Trafficking of APP in the Pathogenesis of Alzheimer's Disease, Front Mol Neurosci 10 (2017) 294.
[8] S. Matsuda, Y. Matsuda, L. D'Adamio, CD74 interacts with APP and suppresses the production of Abeta, Mol Neurodegener 4 (2009) 41.
[9] L.A. Peferoen, D.Y. Vogel, K. Ummenthum, M. Breur, P.D. Heijnen, W.H. Gerritsen, R.M. Peferoen-Baert, P. van der Valk, C.D. Dijkstra, S. Amor, Activation status of human microglia is dependent on lesion formation stage and remyelination in multiple sclerosis, J Neuropathol Exp Neurol 74(1) (2015) 48-63.
[10] J.H. Yun, D.H. Lee, H.S. Jeong, S.H. Kim, S.K. Ye, C.H. Cho, STAT3 activation in microglia increases pericyte apoptosis in diabetic retinas through TNF-É‘/AKT/p70S6 kinase signaling, Biochem Biophys Res Commun 613 (2022) 133-139.

Round 2
Reviewer 1 Report
Comments and Suggestions for Authors
I appreciate the authors' efforts in addressing the previous comments and revising the manuscript. While not all suggestions were fully implemented, the modifications have significantly improved the clarity and rigor of the study. Given the substantial progress made, I recommend acceptance pending minor revisions to address the remaining points below.
- The reference to Figure 1 should be removed from the Abstract, as figures are typically cited only in the main text.
- Figure 17is not referenced in the main text.
- In the figure legend, ‘*, P<0.05; **, P<0.01; ***, P<0.001, t.test’should change to ‘ *Denotes P < 0.05, **P < 0.01 and ***P < 0.001, t-test’.
- Please add a scale bar indicating the 'percent expressed' values to the zoomed-in panels in Figures 10C and 11C.
- In the Materials and Methods section, the authors should describe how the sequencing data were processed to obtain the expression profiles, including what software was used for mapping, whether unique mapped reads were used, the genome version to which the reads were mapped, and what form of expression values were used for subsequent analysis (TPM? Normalized counts? Or RPKM?).
Author Response
Comments 1: The reference to Figure 1 should be removed from the Abstract, as figures are typically cited only in the main text.
Response 1: Thank you for pointing out the issue. We have removed the reference to Figure 1 from the Abstract.
Comments 2: Figure 17 is not referenced in the main text.
Response 2: Thank you for pointing out our issue. We have now correctly referenced Figure 17.
Comments 3: In the figure legend, ‘*, P<0.05; **, P<0.01; ***, P<0.001, t.test’should change to ‘ *Denotes P < 0.05, **P < 0.01 and ***P < 0.001, t-test’.
Response 3: We have replaced ‘*, P<0.05; **, P<0.01; ***, P<0.001, t.test’ with ‘*Denotes P < 0.05, **P < 0.01 and ***P < 0.001, t-test’.
Comments 4: Please add a scale bar indicating the 'percent expressed' values to the zoomed-in panels in Figures 10C and 11C.
Response 4: Thank you for your suggestions on the standardization of our figures. We have now added appropriate scale bars to Figures 10C and 11C.
Comments 5: In the Materials and Methods section, the authors should describe how the sequencing data were processed to obtain the expression profiles, including what software was used for mapping, whether unique mapped reads were used, the genome version to which the reads were mapped, and what form of expression values were used for subsequent analysis (TPM? Normalized counts? Or RPKM?).
Response 5: Thank you for your suggestion to provide a more detailed description of data acquisition and processing. We have now elaborated in the Materials and Methods section on the R package limma used for data processing after acquisition, how probes were mapped to the corresponding genomic chips, the filtering criteria for probes, and the final acquisition of a normalized count matrix.
Thank you once again for your professional advice and your patience.
Reviewer 2 Report
Comments and Suggestions for Authors
1. Figure Clarity:
Several figures, such as Figure 2F, are not clear enough. It is essential to update these figures to ensure that all details are visible and legible.
2. Simplification of the Manuscript:
The manuscript contains a large number of figures, some of which do not directly support the conclusions. It is recommended to simplify the paper by removing less important figures and placing them in the supplementary materials. This will help readers focus on the key findings and improve the paper's conciseness. For example:
Figure 2B-D: These subfigures can be moved to the supplementary materials, as they provide detailed information on the soft threshold selection process, which is less critical for the main narrative.
Figure 4A-E: These subfigures can be moved to the supplementary materials, as they provide detailed information on the machine learning process, which can be referenced but does not need to be prominently displayed in the main text.
Figure 9F: This subfigure can be moved to the supplementary materials, as it provides detailed information on potential drugs targeting specific genes, which is less critical for the main narrative.
3. Consistency Between Text and Figures:
Due to modifications in the figures, some inconsistencies between the text and figures have emerged. It is crucial to ensure that the text accurately reflects the visual data presented in the figures. This includes updating figure captions and references in the text to match the revised figures.
Author Response
Comments 1: Figure Clarity:
Several figures, such as Figure 2F, are not clear enough. It is essential to update these figures to ensure that all details are visible and legible.
Response 1: Thank you for your suggestion on improving the clarity of our figures. We have now replaced Figure 2F with a clearer version.
Comments 2: Simplification of the Manuscript:
The manuscript contains a large number of figures, some of which do not directly support the conclusions. It is recommended to simplify the paper by removing less important figures and placing them in the supplementary materials. This will help readers focus on the key findings and improve the paper's conciseness. For example:
Figure 2B-D: These subfigures can be moved to the supplementary materials, as they provide detailed information on the soft threshold selection process, which is less critical for the main narrative.
Figure 4A-E: These subfigures can be moved to the supplementary materials, as they provide detailed information on the machine learning process, which can be referenced but does not need to be prominently displayed in the main text.
Figure 9F: This subfigure can be moved to the supplementary materials, as it provides detailed information on potential drugs targeting specific genes, which is less critical for the main narrative.
Response 2: Thank you for pointing out the redundancy in some of our figures. We have now moved Figures 2 B-D, G; 4A, B, E; and 9F to the supplementary file.
Comments 3: Consistency Between Text and Figures:
Due to modifications in the figures, some inconsistencies between the text and figures have emerged. It is crucial to ensure that the text accurately reflects the visual data presented in the figures. This includes updating figure captions and references in the text to match the revised figures.
Response 3: Thank you for reminding us to match the figures with the main text. We have now checked the correspondence between the figures and the main text, and after making the necessary revisions, we have checked again. We have also checked the correspondence between the figure captions and the figure contents and made the necessary corrections.
Thank you once again for your professional advice and your patience.
Reviewer 3 Report
Comments and Suggestions for Authors
Comments to the Authors,
It should be possible to perform the analysis using immunohistological techniques in the EAE model without MS patients, which is a fundamental part of their argument in this paper, and it seems imperative to present that data. If they cannot show it, then the title of the paper should be changed or the content should be changed more.
Author Response
Comments 1:
Comments to the Authors, It should be possible to perform the analysis using immunohistological techniques in the EAE model without MS patients, which is a fundamental part of their argument in this paper, and it seems imperative to present that data. If they cannot show it, then the title of the paper should be changed or the content should be changed more.
Response 1: Thank you very much for your constructive suggestion. We fully agree that validating the spatial expression of APP and CD74 in the EAE model is essential for strengthening our hypothesis. In response, we have now performed immunohistochemistry (IHC) and immunofluorescence (IF) staining in EAE mouse brain tissues.
Specifically, Supplementary Figure 5 shows the IHC results for APP and CD74 in both normal and inflamed brain tissue. Supplementary Figure 6 further provides IF co-localization data demonstrating that APP is predominantly expressed in pericytes (PDGFRβ^+) and CD74 in microglia (IBA1^+), supporting the ligand–receptor relationship inferred from our single-cell analysis.
We have also revised the Discussion and Conclusion sections to more rigorously state the limitations of our data and explicitly indicate that these findings serve as a hypothesis-generating basis for future functional studies. We thank the reviewer for highlighting this important point, which has helped significantly improve the robustness of our manuscript.

Round 3
Reviewer 3 Report
Comments and Suggestions for Authors accept in the current form